# Somatic hypermutation of T cell receptor α chain contributes to selection in nurse shark thymus

Jeannine A Ott[1], Caitlin D Castro[2], Thaddeus C Deiss[1], Yuko Ohta[2], Martin F Flajnik[2], Michael F Criscitiello[1,3]*

[1]Comparative Immunogenetics Laboratory, Department of Veterinary Pathobiology, College of Veterinary Medicine and Biomedical Sciences, Texas A&M University, Texas, United States; [2]Department of Microbiology and Immunology, University of Maryland at Baltimore, Baltimore, United States; [3]Department of Microbial Pathogenesis and Immunology, College of Medicine, Texas A&M Health Science Center, Texas A&M University, Texas, United States

**Abstract** Since the discovery of the T cell receptor (TcR), immunologists have assigned somatic hypermutation (SHM) as a mechanism employed solely by B cells to diversify their antigen receptors. Remarkably, we found SHM acting in the thymus on α chain locus of shark TcR. SHM in developing shark T cells likely is catalyzed by activation-induced cytidine deaminase (AID) and results in both point and tandem mutations that accumulate non-conservative amino acid replacements within complementarity-determining regions (CDRs). Mutation frequency at TcRα was as high as that seen at B cell receptor loci (BcR) in sharks and mammals, and the mechanism of SHM shares unique characteristics first detected at shark BcR loci. Additionally, fluorescence in situ hybridization showed the strongest AID expression in thymic corticomedullary junction and medulla. We suggest that TcRα utilizes SHM to broaden diversification of the primary αβ T cell repertoire in sharks, the first reported use in vertebrates.
DOI: https://doi.org/10.7554/eLife.28477.001

*For correspondence:
mcriscitiello@cvm.tamu.edu

Competing interests: The authors declare that no competing interests exist.

## Introduction

All jawed vertebrates share fundamental components of the adaptive immune system. The cartilaginous fish (including sharks) are the most divergent jawed vertebrate group relative to mammals and use a polymorphic major histocompatibility complex (MHC) (*Kasahara et al., 1992*), multiple isotypes of immunoglobulin (Ig) heavy and light chains (*Flajnik, 2002*; *Criscitiello and Flajnik, 2007*), and the typical four T-cell receptor (TcR) chains (*Rast et al., 1997*). Shark lymphocyte antigen receptors are diversified by RAG-mediated V(D)J somatic rearrangement (*Bernstein et al., 1994*). After antigen exposure, B cells also use the enzyme activation-induced cytidine deaminase (AID) for receptor modification via somatic hypermutation (SHM) (*Conticello et al., 2005*), allowing activated B cells to extensively alter their rearranged Ig variable region genes (*Muramatsu et al., 2000*). Some of the variants produced by this process bind antigen with higher affinity, enhancing humoral immunity through affinity maturation. In addition to SHM in all jawed vertebrate Igs, AID catalyzes the processes of heavy chain class switch recombination (CSR) in tetrapods (and is implicated in shark CSR [*Zhu et al., 2012*]) and Ig gene conversion (in birds and some mammals) (*Barreto and Magor, 2011*). AID is a member of the APOBEC family of nucleic acid mutators, two of which likely diversify the variable lymphocyte receptor (VLR) system in the more ancient vertebrate lineages of lamprey and hagfish (*Alder et al., 2005*; *Guo et al., 2009*).

Although in general immunologists think TcR loci do not undergo somatic hypermutation, a few reports do exist of AID-mediated SHM in T cells. However, non-productive TcRα rearrangements in hybridomas (*Marshall et al., 1999*), TcRβ sequences from HIV-positive individuals (*Cheynier et al., 1998*), and reports of SHM in TcRα murine germinal center T cells (*Zheng et al., 1994*) are not thought to describe any normal physiology (*Bachl and Wabl, 1995*). More recent studies indicate that TcR δ and γ in the dromedary camel and TcR γ in the sandbar shark somatically hypermutate (see below) (*Antonacci et al., 2011*; *Chen et al., 2012*). Despite these findings, the general consensus has remained that AID does not target TcR loci (*Pavri and Nussenzweig, 2011*; *Choudhary et al., 2018*). In over 30 years of studies of TcR repertoires, it has been clear that SHM is not functioning to either generate or further enhance the TcR repertoire of mouse and human.

Recent studies in the sandbar shark (*Carcharhinus plumbeus*) revived the notion of SHM at TcR loci. Sequencing of the entire TcRγ translocon in *C. plumbeus* showed definitively that SHM is occurring at that locus (*Chen et al., 2009*). Shark TcRγ SHM occurs in two distinct patterns: point mutations and tandem mutations characteristic of B cell SHM in cartilaginous fish (*Anderson et al., 1995*; *Lee et al., 2002*; *Rumfelt et al., 2002*; *Zhu et al., 2012*), possibly suggesting two different cellular mechanisms for generating mutations (*Chen et al., 2012*). The sandbar shark analysis found targeted nucleotide motifs of AID activity at the TcRγ locus. *Chen et al. (2012)* examined ratios of replacement (R) and silent (S) mutations between CDR and framework regions to determine if mutation altered affinity of receptors, a method commonly used to study B cell affinity maturation by SHM. Finding no difference between R/S ratios in CDR versus framework regions, they concluded that TcRγ uses SHM to generate a more diverse repertoire rather than for affinity maturation. SHM-induced changes to TcRδ in camels showed similar results.

Early work in our lab also suggested that SHM occurs in the less restricted γδ T cells in nurse shark (*Ginglymostoma cirratum*) and perhaps in the alpha chain of MHC-restricted αβ T cells (*Criscitiello et al., 2010*), encouraging us to examine this phenomenon further. Thus, we performed a systematic analysis of this process in shark primary and secondary lymphoid tissues using thymocyte clones containing the same unique third complementarity-determining region (CDR3). Our data suggest that SHM of TcRα is involved in primary T cell repertoire diversification and the enhancement of positive or negative selection in the thymic cortex. This finding is consistent with a model put forward by Niels Jerne over 40 years ago to explain antigen receptor positive selection (*Jerne, 1971*).

## Results

### Somatic hypermutation in TcR γV and TcR δV

We assessed the presence of possible SHM within TcR V segments from γ, δ, and β chains. Using neighbor-joining consensus trees, we grouped sequences from each chain into V families based on 85% nucleotide identity and into V subfamilies based on 90% nucleotide identity. We then examined each V subfamily for the possibility of SHM. However, since none of the sequences from these chains contained CDR3 regions, we did not rigorously analyze mutations within these chains.

We first corroborated the original finding of SHM in TcRγ variable regions (V) in sandbar shark spleen (*Chen et al., 2012*; *Chen et al., 2009*) using peripheral lymphoid tissue from the spiral valve (intestine) of nurse shark. We also examined mutation in clones from the thymus. We conservatively assigned 69 sequences to 9 V genes from three TcRγ families (*Figure 1*). Even with a conservative assignment of clones to predicted germline V sequences, we found nearly twice as many V genes as in the sandbar shark locus (which only contains five Vs). Only TcR γ4 did not display mutations in the nurse shark, but since we found only two γ4 sequences, it is possible that it occurs but our sample was too small to observe it.

We then investigated the possibility of mutation occurring at the TcRδ locus. We analyzed mutation in clones from nurse shark thymus, peripheral blood leukocytes, and spiral valve, conservatively grouping 111 clones into 12 V genes from 7 TcRδ families (*Figure 2*). Only one of these families (TcR δ10) lacked mutation. We found that in five of the seven δV families, the same V-δ segments used to generate δ cDNA sequences also generated α cDNA sequences. The sequence diversity at TcRγ and TcRδ was in contrast to TcRβ, where we found no such evidence for mutation in 56 sequences representing six V segments from six different V families (*Figure 3*). Limited existing data also do not

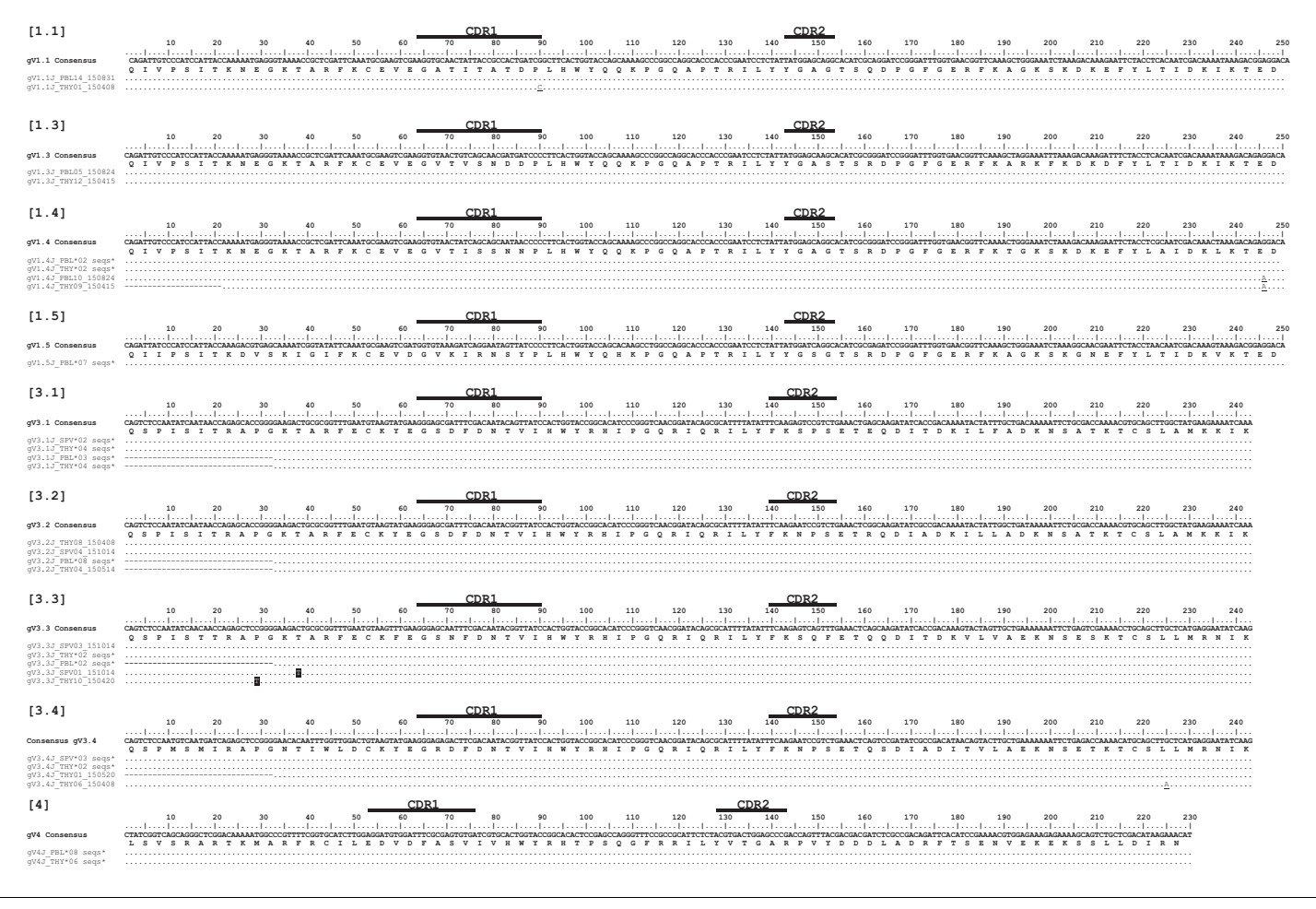

**Figure 1.** Alignment of Gamma V clones suggests minimal somatic hypermutation. Thymocyte clones for nine γV groups from three different predicted V genes. CDR regions are marked above the scale for each γV alignment. Amino acids are shown under the nucleotide consensus sequence, and dots represent identity to this sequence. We highlighted nonsynonymous changes in black; synonymous changes are underlined. Gaps are used for alignment purposes and indicate a shortened sequence (at the beginning or end). Sequences are identified by a single clone number or a group of identical clones condensed to a single line (the number of clones are indicated). Clone numbers that contain 'THY' are from thymus, 'PBL' are from peripheral blood leukocytes, and 'SPV' are from spiral valve (intestine). We deposited all 69 sequences into GenBank under accession numbers KY351639 – KY351707.

DOI: https://doi.org/10.7554/eLife.28477.002

support mutation at NAR-TcR, a distinct TcR containing a NAR V domain supported by a more canonical V δ domain, each resulting from independent V(D)J rearrangements (*Criscitiello et al., 2006*) (data not shown).

## Identification of TcR Vα genes in the nurse shark genome

We identified 17 germline α/δ V gene sequences corresponding to 12 unique V segments. Unfortunately, these segments matched only two groups of sequences in our TcRα dataset (TcRα/δ V4 and V9), likely due to inter-individual polymorphisms. In the absence of a complete germline sequence for this locus, we limited our database for TcRα to thymocyte clones with the same unique CDR3 signature. The CDR3 region results from the somatic recombination and assembly of variable (V) and joining (J) gene segments during lymphocyte development in the thymus (*Kuklina, 2006*; *Lantelme et al., 2008*). The recombination process cleaves DNA and initiates repair mechanisms that result in the random insertion of non-template (N) nucleotides within the join, forming a unique binding sequence that contributes to the diversity and specificity of a TcR (*Gellert, 2002*;

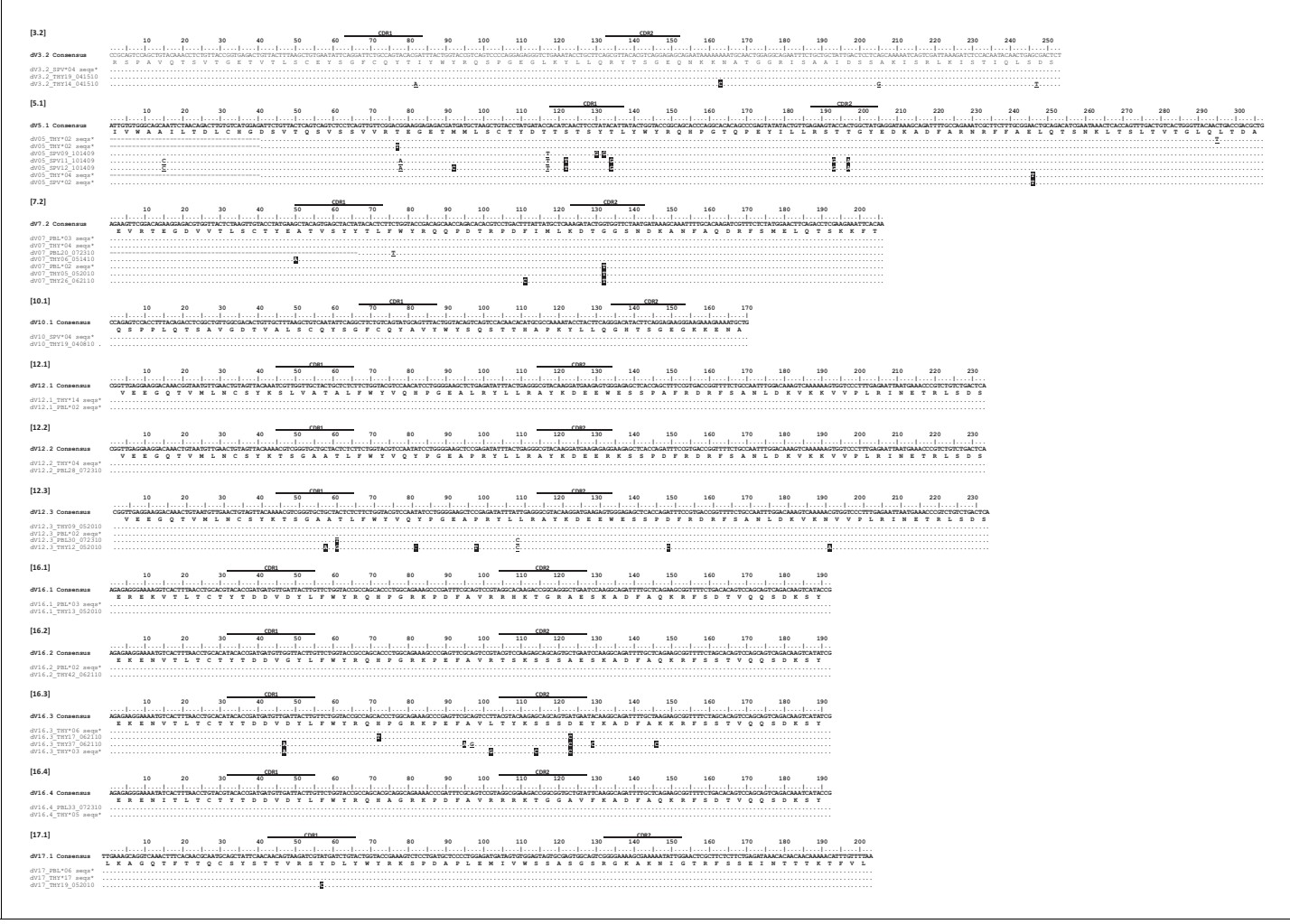

**Figure 2.** Alignment of Delta V clones suggests somatic hypermutation. Thymocyte clones for 12 δV groups from seven different predicted V genes. CDR regions are marked above the scale for each δV alignment. Amino acids are shown under the nucleotide consensus sequence, and dots represent identity to this sequence. We highlighted nonsynonymous changes in black; synonymous changes are underlined. Gaps are used for alignment purposes and indicate a shortened sequence (at the beginning or end). Sequences are identified by a single clone number or a group of identical clones condensed to a single line (the number of clones are indicated). Clone numbers that contain 'THY' are from thymus, 'PBL' are from peripheral blood leukocytes, and 'SPV' are from spiral valve (intestine). We deposited all 112 sequences into GenBank under accession numbers KY346705 – KY346816.

DOI: https://doi.org/10.7554/eLife.28477.003

*Kuklina, 2006*). Once recombination ceases and the thymocyte proliferates, this distinctive CDR3 sequence is perpetuated in all daughter cells (*Murphy and Weaver, 2017*). Since it is unlikely that two thymocytes would generate identical CDR3 sequences during VJ recombination, we predict that amplicons containing identical CDR3 sequences derived from the same progenitor and thus must contain the same germline V and J segments. Alpha CDR3s exhibited substantial variation within our shark sequences, despite the absence of diversity (D) segments. For example, TcRα V1 sequences using the same V and J segments had CDR3 lengths that differed by as many as six amino acids (18 nucleotides), and few CDR3s shared more than one amino acid within this V-J join (*Figure 4*). Further, the majority of sequences in our overall dataset do contain N and palindromic (P) nucleotides within this join. Using clones containing the same V segment from different sharks, we determined the putative end of each V segment. We first aligned sequences containing the same V segment. Then, assuming that any nucleotide present in the same position within more than one shark must be germline, we determined which nucleotides within a join belong to the V segment. We then

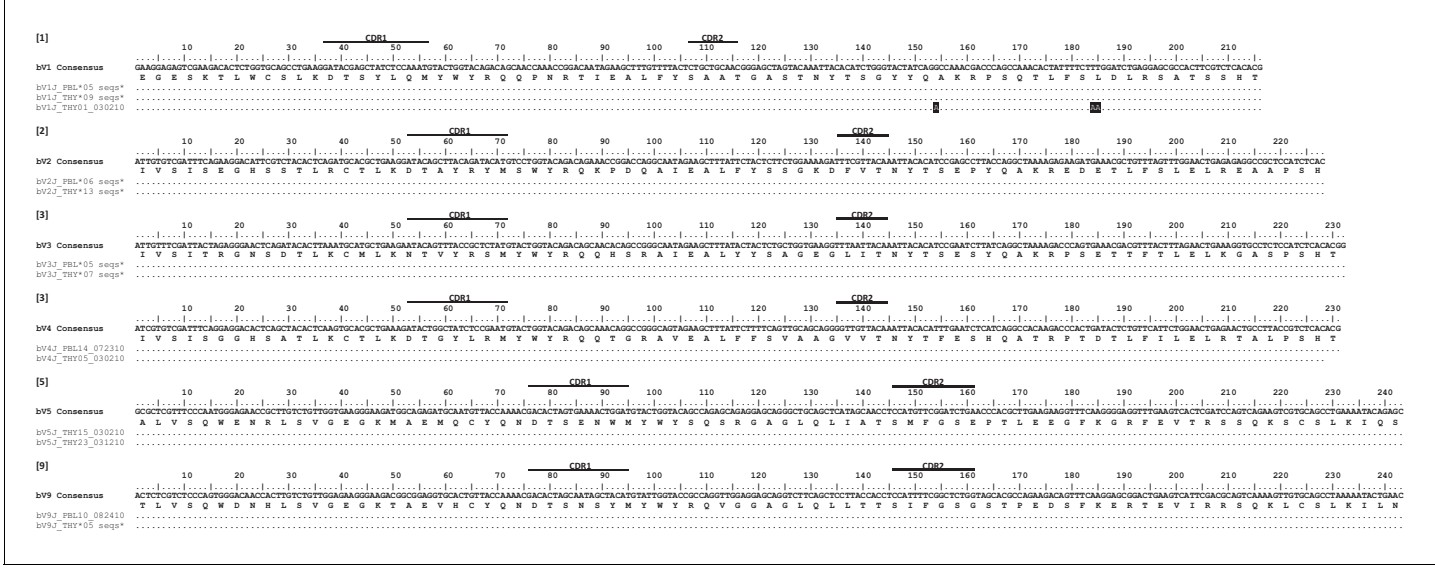

**Figure 3.** Alignment of Beta V clones illustrates a lack of somatic hypermutation. Thymocyte clones for βV groups from six different predicted V genes. CDR regions are marked above the scale for each βV alignment. Amino acids are shown under the nucleotide consensus sequence, and dots represent identity to this sequence. We observed three nonsynonymous changes within a single sequence (highlighted in black). Sequences are identified by a single clone number or a group of identical clones condensed to a single line (the number of clones are indicated). Clone numbers that contain 'PBL' are from peripheral blood leukocytes and 'THY' are from thymus. We deposited all 57 sequences into GenBank under accession numbers KY351708 – KY351764.

DOI: https://doi.org/10.7554/eLife.28477.004

repeated this process for each J segment. Of 290 clones, we found 197 (68%) unique sequences within this join (0–34 nucleotides in length), suggesting that most sequences contain N/P nucleotides

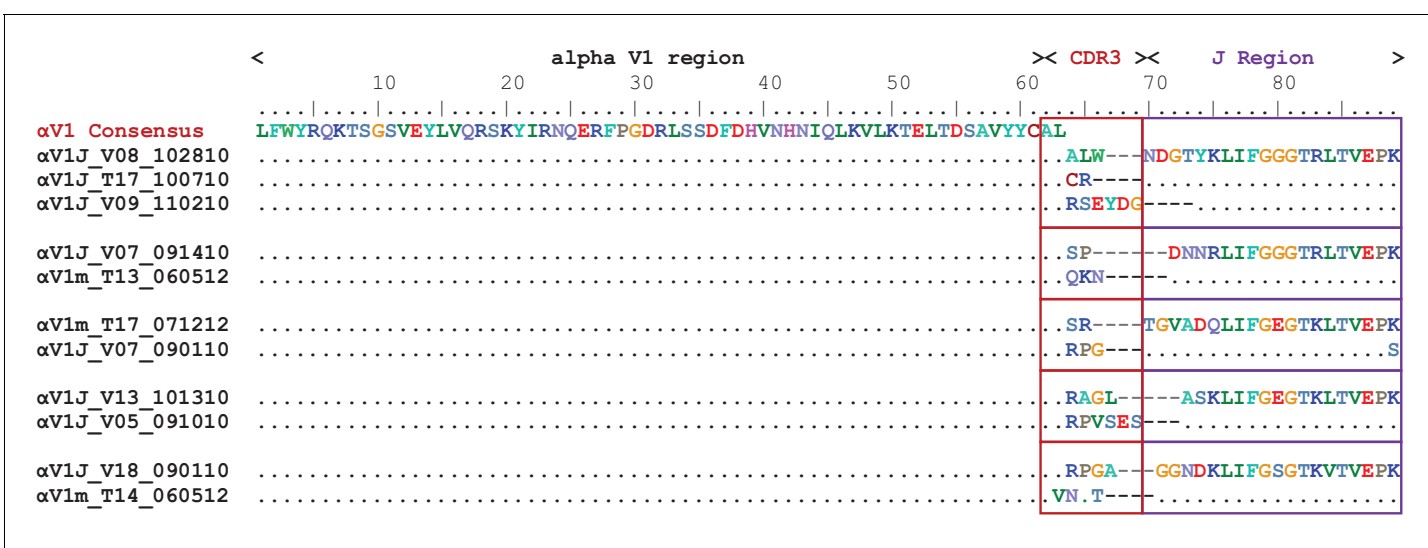

**Figure 4.** CDR3s of TcR Alpha chain are diverse. Amino acid (aa) alignment of TcR αV1 thymocyte clones illustrating diversity of the third complementarity-determining region (CDR3). All clones contain identical variable (V) region sequence (aa 1–61). We grouped clones by shared, identical joining (J) regions (purple boxes) and highlight the differences in the V-J join (CDR3 region) in red boxes.

DOI: https://doi.org/10.7554/eLife.28477.005

The following source data is available for figure 4:

**Source data 1.** CDR3 regions diversified by exonuclease activity and addition of N and P nucleotides.

DOI: https://doi.org/10.7554/eLife.28477.006

**Table 1.** Summary of sequence data used in this paper.
Putative subfamilies within each TcR alpha V family share at least 85% nucleotide identity using nearest-neighbor consensus trees of V segments. Number of TcR alpha nucleotide (NUC), amino acid (AA) sequences or sequence groups within each category. Highlighted columns specifically refer to data used in this study. (See results for detailed descriptions of sequences included within each column.)

| Alpha V segment | Putative # Sub-families | All cloned sequences | Complete CDR3-J junction[*] | Unique V Region [†] | | Unique V segment[‡] | | Unique CDR3-J[§] | | Groups with identical CDR3-J[#] | CDR3-J groups in Study[**] | Sequences in each dataset[††] |
|---|---|---|---|---|---|---|---|---|---|---|---|---|
| | | | | NUC | AA | NUC | AA | NUC | AA | | | |
| TRA V1 | 1 | 40 | 40 | 35 | 34 | 24 | 16 | 34 | 32 | 7 | 4 | 2, 3, 3, 5 |
| TRA V2 | 3 | 18 | 18 | 13 | 13 | 13 | 10 | 12 | 12 | 3 | 1 | 5 |
| TRA V3 | 3 | 217 | 194 | 55 | 52 | 35 | 34 | 51 | 50 | 5 | 1 | 4 |
| TRA V4 | 3 | 60 | 28 | 22 | 22 | 21 | 21 | 21 | 21 | 6 | 2 | 2, 2 |
| TRA V5 | 4 | 35 | 34 | 15 | 13 | 9 | 8 | 13 | 13 | 3 | 1 | 2 |
| TRA V6 | 2 | 9 | 9 | 7 | 7 | 5 | 6 | 7 | 7 | 1 | 0 | 0 |
| TRA V7 | 5 | 96 | 60 | 49 | 48 | 39 | 38 | 48 | 48 | 9 | 2 | 2, 2 |
| TRA V9 | 3 | 19 | 19 | 14 | 14 | 12 | 11 | 13 | 13 | 4 | 0 | 0 |
| TRA V10 | 2 | 45 | 45 | 29 | 26 | 21 | 19 | 27 | 26 | 10 | 2 | 4, 9 |
| | | 539 | 447 | 239 | 229 | 179 | 163 | 226 | 224 | 48 | **13** | **45** |

[*]A full list of these sequences can be found in **Table 1—source data 1**.

[†]**V Region** includes all bases between the 1 st predicted nucleotide of the V segment to the last predicted nucleotide of the J segment (V and J).

[‡]**V Segment** includes all bases between the 1 st predicted nucleotide of the V segment to the last predicted nucleotide of the V segment (V only).

[§]**CDR3-J** includes all bases after the last predicted nucleotide of the V segment to the last predicted nucleotide of the J segment.

[#]Number of groups with identical CDR3-J sequences, which we used to determine sequence relatedness (see text for details).

[**]Number of groups with identical CDR3-J sequences used in this study. (Those not used contained no mutation with V segments.)

[††]Total number of sequences for each alpha V used to assess somatic hypermutation within this study (e.g., for aV1, 4 different clonal groups contained 2, 3, 3, and 5 identical CDR3-J regions, respectively).

DOI: https://doi.org/10.7554/eLife.28477.007

The following source data available for Table 1:
Source data 1. Tab-delimited text file containing sequences used in this paper for analysis of somatic hypermutation of TcR alpha chain.
DOI: https://doi.org/10.7554/eLife.28477.008

(see *Figure 4—source data 1*). Finally, we never observed the same CDR3 sequence in more than one shark, suggesting both exonuclease activity and addition of N and P nucleotides help diversify alpha CDR3s in nurse shark. Therefore, even in the absence of an assembled locus, we were able to evaluate mutation to germline αV segments by considering changes within only those thymocyte clones containing identical CDR3s. Our extremely conservative approach of using *only clones containing the same CDR3 encoding rearrangement and N and P nucleotide sequences* provided us the assurance that we had distinct αVs descendant from clonal T cells, since it would be extremely unlikely that two T cells created receptors that contained the exact same nucleotide sequence by chance.

## Somatic hypermutation in nurse shark TcR αV

With SHM confirmed in γ and δ TcR chains but apparently not the TcR beta chain of nurse shark, we checked for mutation of the TcRα locus. One might expect mutation in γδ T cells since antigen binding more closely mirrors that of B cells. However, mutations to receptors of MHC-restricted αβ T cells would be surprising given that even minor modifications to these receptors could risk incompatibility with MHC.

Our preliminary V α dataset contained 539 TcRα clones (encoding 286 unique amino acid sequences representing nine V α families) from three tissues (PBL, spleen, thymus) of two sharks (*Joanie, Mary Junior*). Of this total, only 447 sequences contained complete CDR3-J junctions (including all bases after the last predicted nucleotide of the V segment to the last predicted nucleotide of the J segment; see *Table 1*). We observed 239 (53.5%) sequences with unique V regions (from the first predicted nucleotide of the V segment to the last predicted nucleotide of the J segment) and 179

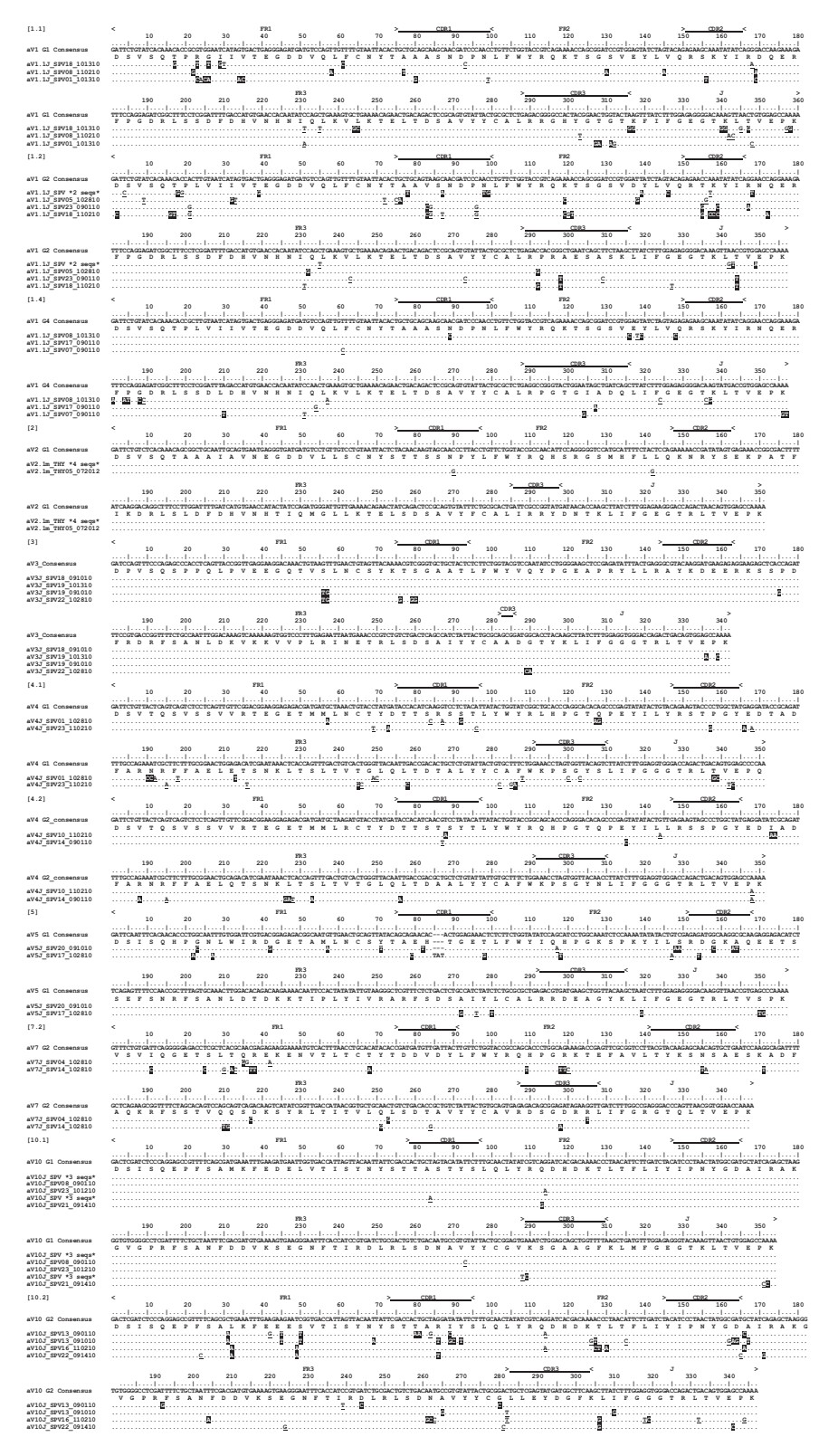

**Figure 5.** Alignment of Alpha V cDNA clones suggest somatic hypermutation at shark TCRα. Thymocyte clones for all 11 αV groups with the same CDR3 from six different predicted V genes. Locations of framework regions (FR), complementarity determining regions (CDR), joining regions (J), and constant (C) regions are marked above the scale for each αV. In absence of germline sequence information, we used a Geneious-derived nucleotide consensus sequence for analysis of nucleotide changes in thymocyte clones. Amino acids are shown under the consensus sequence, and dots represent

*Figure 5 continued on next page*

*Figure 5 continued*

identity to this sequence. Nonsynonymous changes are highlighted in black; synonymous changes are underlined. Gaps are used for alignment purposes and indicate either a shortened sequence (at the beginning or end) or insertions or deletions within the sequence. Sequences are identified by a single clone number or a group of identical clones condensed to a single line (the number of clones are indicated). Clone numbers that contain 'THY' are from thymus, 'PBL' are from peripheral blood leukocytes, and 'SPV' are from spiral valve (intestine). We did not use clones from αV1.3 and αV7.1 because they did not contain mutations in FR or CDR regions. We deposited all 42 sequences into GenBank under accession numbers KY189332 – KY189354 or KY366469 – KY366487.

DOI: https://doi.org/10.7554/eLife.28477.009

(40%) with unique V segments (from the first to the last predicted nucleotide of the V segment only). We found 226 sequences (50%) containing unique CDR3-J regions (all bases after the last predicted nucleotide of the V to the last predicted nucleotide of the J). However, we found 48 groups containing *identical* CDR3-J sequences across all nine V α families (suggesting they bear the V-J rearrangement from a single founder thymocyte), each V α family containing anywhere from one to ten clonal groups (*Table 1*). The majority of these groups contained no mutation within V, J or C regions. For example, one α V3 sequence group occurred 131 times, the most numerous sequence in the dataset, yet contained no mutation in any sequence. We did observe mutation in 12 of these 48 groups belonging to seven different V α families. Each family contained between one and four clonal groups, and each group contained two to nine sequences with identical CDR3-J regions (45 total sequences; see *Table 1*). We include these 45 sequences in our TcR V α dataset (see *Table 1* – source data for sequence data).

Using these 45 sequences from both thymus and peripheral immune tissues, we found evidence for SHM acting on the TcRα genes. We divided our 45 clones into 13 CDR3-sharing groups from seven different α V families and then analyzed sequences within groups for potential mutation (*Figure 5*). We excluded two groups (four sequences) from analyses (αV1.3 and αV7.1) that contained no mutations within FR or CDR regions (leaving 41 clones for analysis). Two sequences (*aV7.4m_-THY09_051410* and *aV5J_SPV17_102810*) contained one 3-base insertion and one sequence (*aV1.4J_SPV07_090110*) contained an 18-base insertion; although SHM can result in insertions and deletions (*Diaz et al., 2002*), we did not include insertion nucleotides in mutation counts. All sequences were in-frame and contained no internal stop codons, suggesting functionality of cells. Average lengths of CDRs were as follows: CDR1: 7.0 amino acids (range: 5–8); CDR2: 5.7 amino acids (range: 5–7); and CDR3: 6.0 amino acids (range: 1–10). Naming of families and subfamilies followed *Criscitiello et al. (2010)*. However, with the accumulation of sequence data over our previous analysis of nurse shark TcRα (*Criscitiello et al., 2010*), we expanded our nomenclature considerably. Additional annotation followed the IMGT guidelines for TcRs (*Lefranc et al., 2003*).

*Figure 5* shows all 12 αV CDR3 groups exhibiting mutation. The overall TcRα mutation frequency was 0.0226 substitutions per nucleotide (S/N), with 66% of all substitutions (187 of 283) resulting in amino acid replacements. The CDRs accumulated significantly more mutations than FRs (CDR: 0.0352 S/N; FR: 0.0188 S/N; p=0.0373), and substitutions in CDRs were twice as likely to be nonsynonymous changes (NSYN) than those in FRs (CDR: 0.0235 S/N; FR: 0.0122 S/N; p=0.0312; *Table 2*). There was no difference in frequency of synonymous (SYN) mutations between regions (CDR: 0.0117 S/N, FR: 0.0066 S/N; p=0.0705). Finally, although we found more tandemly mutated bases in CDRs (41 of 81, or 50.6% of all CDR mutations) than in FRs (73 of 192, or 38.0% of all FR mutations), this difference was not significant; p=0.721; *Table 3*). Tandem mutations ranged from two to four bases in length (mean = 2.78 bases). That this feature of SHM, specific of cartilaginous fish Ig (*Rumfelt et al., 2001*; *Lee et al., 2002*; *Diaz et al., 1999*), also occurs in the TcR strongly supports the validity of our analyses.

Mutation frequency also varied by region (CDR or FR). The highest mutation frequency occurred in CDR1 and accumulated significantly more mutations overall than other regions ($\bar{X}$=4.48%; see *Figures 6* and *7a*, *Table 2*). CDR3 displayed an unusually low mean mutation frequency (2.97%). However, these frequencies may be artificially low since our groups were based only on clones containing the same CDR3 sequence, and clones whose CDR3s deviated markedly from the consensus would have been excluded by our conservative grouping approach. We observed the lowest mean mutation frequency in FR3 (1.64%). These results are consistent with what is known about human TcR binding to MHC:Ag structures. CDR1 and CDR3 make more contacts with Ag while

**Table 2.** Frequencies of somatic hypermutation in nurse shark alpha V groups (αV G) containing the same CDR3. Mutation frequency is the total number of nucleotide changes to a Geneious-derived consensus sequence divided by the total number of nucleotides. We counted synonymous (S) and nonsynymous (N) mutations separately for each FR and CDR for 11 different CDR3 groups in seven predicted alpha V genes. [FR: framework region; CDR: complementarity-determining region; Seqs: sequences; Nuc: nucleotides; Freq: frequency]

**FR1**

| αV Group | # Seqs | # Codons | Total Nuc | Mutations S | Mutations N | Total Mutation | Mutation Freq (%) |
|---|---|---|---|---|---|---|---|
| αV1.1 | 3 | 25 | 225 | 2 | 12 | 14 | 6.222 |
| αV1.2 | 5 | 25 | 375 | 10 | 8 | 18 | 4.800 |
| αV1.4 | 3 | 25 | 225 | 1 | 0 | 1 | 0.444 |
| αV2.1 | 5 | 25 | 375 | 0 | 0 | 0 | 0.000 |
| αV3.1 | 4 | 25 | 300 | 0 | 4 | 4 | 1.333 |
| αV4.1 | 2 | 25 | 150 | 1 | 2 | 3 | 2.000 |
| αV4.2 | 2 | 25 | 150 | 0 | 0 | 0 | 0.000 |
| αV5.1 | 2 | 25 | 150 | 0 | 6 | 6 | 4.000 |
| αV7.2 | 2 | 25 | 150 | 4 | 7 | 11 | 7.333 |
| αV10.1 | 4 | 25 | 300 | 0 | 0 | 0 | 0.000 |
| αV10.2 | 9 | 25 | 675 | 2 | 11 | 13 | 1.926 |
| Sum | 41 | 275 | 3075 | 20 | 50 | 70 | |
| Mean Mutation Freq (%) | | | | 0.65 | 1.63 | | 2.28 |
| Standard Deviation | | | | 2.994 | 4.547 | 6.531 | 2.979 |

**FR2**

| αV Group | # Codons | Total Nuc | Mutations S | Mutations N | Total Mutation | Mutation Freq (%) |
|---|---|---|---|---|---|---|
| αV1.1 | 17 | 153 | 0 | 2 | 2 | 1.307 |
| αV1.2 | 17 | 255 | 1 | 8 | 9 | 3.529 |
| αV1.4 | 17 | 153 | 1 | 3 | 4 | 2.614 |
| αV2.1 | 17 | 255 | 1 | 0 | 1 | 0.392 |
| αV3.1 | 17 | 204 | 0 | 0 | 0 | 0.000 |
| αV4.1 | 17 | 102 | 0 | 2 | 2 | 1.961 |
| αV4.2 | 17 | 102 | 1 | 1 | 2 | 1.961 |
| αV5.1 | 17 | 102 | 2 | 3 | 5 | 4.902 |
| αV7.2 | 17 | 102 | 1 | 3 | 4 | 3.922 |
| αV10.1 | 17 | 204 | 1 | 1 | 2 | 0.980 |
| αV10.2 | 17 | 459 | 4 | 4 | 8 | 1.743 |
| Sum | 187 | 2091 | 12 | 27 | 39 | |
| Mean Mutation Freq (%) | | | 0.57 | 1.29 | | 1.87 |
| Standard Deviation | | | 1.136 | 2.252 | 2.841 | 2.248 |

**FR3**

| αV Group | # Codons | Total Nuc | Mutations S | Mutations N | Total Mutation | Mutation Freq (%) |
|---|---|---|---|---|---|---|
| αV1.1 | 41 | 369 | 4 | 4 | 8 | 2.168 |
| αV1.2 | 41 | 615 | 5 | 5 | 10 | 1.626 |
| αV1.4 | 41 | 369 | 4 | 5 | 9 | 2.439 |
| αV2.1 | 42 | 630 | 0 | 0 | 0 | 0.000 |
| αV3.1 | 40 | 480 | 0 | 1 | 1 | 0.208 |
| αV4.1 | 43 | 258 | 11 | 7 | 18 | 6.977 |
| αV4.2 | 43 | 258 | 2 | 7 | 9 | 3.488 |
| αV5.1 | 42 | 252 | 1 | 2 | 3 | 1.190 |
| αV7.2 | 41 | 246 | 1 | 6 | 7 | 2.846 |
| αV10.1 | 41 | 492 | 4 | 0 | 4 | 0.813 |
| αV10.2 | 40 | 1080 | 3 | 11 | 14 | 1.296 |
| Sum | 455 | 5049 | 35 | 48 | 83 | |
| Mean Mutation Freq (%) | | | 0.69 | 0.95 | | 1.64 |
| Standard Deviation | | | 3.125 | 3.414 | 5.429 | |

**FR Means**

| αV Group | S | N | ALL |
|---|---|---|---|
| αV1.1 | 6 | 18 | 24 |
| αV1.2 | 16 | 21 | 37 |
| αV1.4 | 6 | 8 | 14 |
| αV2.1 | 1 | 0 | 1 |
| αV3.1 | 0 | 5 | 5 |
| αV4.1 | 12 | 11 | 23 |
| αV4.2 | 3 | 8 | 11 |
| αV5.1 | 3 | 11 | 14 |
| αV7.2 | 6 | 16 | 22 |
| αV10.1 | 5 | 1 | 6 |
| αV10.2 | 9 | 26 | 35 |
| Sum | 67 | 125 | 192 |
| Mean Mutation Freq (%) | 0.66 | 1.22 | 1.88 |
| Standard Deviation | 4.742 | 8.201 | 11.861 |

**CDR1**

| AlphaV Group | # Seqs | # Codons | Total Nuc | Mutations S | Mutations N | Total Mutation | Mutation Freq (%) |
|---|---|---|---|---|---|---|---|
| αV1.1 | 3 | 8 | 72 | 2 | 2 | 4 | 5.556 |
| αV1.2 | 5 | 8 | 120 | 5 | 4 | 9 | 7.500 |
| αV1.4 | 3 | 8 | 72 | 0 | 1 | 1 | 1.389 |
| αV2.1 | 5 | 7 | 105 | 0 | 1 | 1 | 0.952 |
| αV3.1 | 4 | 6 | 72 | 0 | 3 | 3 | 4.167 |
| αV4.1 | 2 | 7 | 42 | 3 | 1 | 4 | 9.524 |
| αV4.2 | 2 | 7 | 42 | 1 | 1 | 2 | 4.762 |
| αV5.1 | 2 | 7 | 42 | 0 | 3 | 3 | 7.143 |
| αV7.2 | 2 | 5 | 30 | 0 | 0 | 0 | 0.000 |
| αV10.1 | 4 | 7 | 84 | 3 | 0 | 3 | 3.571 |
| αV10.2 | 9 | 7 | 189 | 1 | 8 | 9 | 4.762 |
| Sum | 41 | 77 | 870 | 15 | 24 | 39 | |
| Mean Mutation Freq (%) | | | | 1.72 | 2.76 | | 4.48 |
| Standard Deviation | | | | 1.690 | 2.316 | 2.979 | |

**CDR2**

| αV Group | # Codons | Total Nuc | Mutations S | Mutations N | Total Mutation | Mutation Freq (%) |
|---|---|---|---|---|---|---|
| αV1.1 | 5 | 45 | 1 | 0 | 1 | 2.222 |
| αV1.2 | 5 | 75 | 1 | 2 | 3 | 4.000 |
| αV1.4 | 5 | 45 | 0 | 0 | 0 | 0.000 |
| αV2.1 | 5 | 75 | 0 | 0 | 0 | 0.000 |
| αV3.1 | 6 | 72 | 0 | 0 | 0 | 0.000 |
| αV4.1 | 6 | 36 | 0 | 1 | 1 | 2.778 |
| αV4.2 | 6 | 36 | 0 | 0 | 0 | 0.000 |
| αV5.1 | 6 | 36 | 0 | 7 | 7 | 19.444 |
| αV7.2 | 7 | 42 | 1 | 1 | 2 | 4.762 |
| αV10.1 | 6 | 36 | 0 | 0 | 0 | 0.000 |
| αV10.2 | 6 | 162 | 2 | 2 | 4 | 2.469 |
| Sum | 63 | 696 | 5 | 13 | 18 | |
| Mean Mutation Freq (%) | | | 0.72 | 1.87 | | 2.59 |
| Standard Deviation | | | 0.688 | 2.089 | 2.248 | |

**CDR3**

| αV Group | # Codons | Total Nuc | Mutations S | Mutations N | Total Mutation | Mutation Freq (%) |
|---|---|---|---|---|---|---|
| αV1.1 | 10 | 90 | 1 | 6 | 7 | 7.778 |
| αV1.2 | 8 | 120 | 1 | 4 | 5 | 4.167 |
| αV1.4 | 5 | 45 | 0 | 2 | 2 | 4.444 |
| αV2.1 | 3 | 45 | 0 | 0 | 0 | 0.000 |
| αV3.1 | 1 | 12 | 0 | 0 | 0 | 0.000 |
| αV4.1 | 6 | 36 | 2 | 0 | 2 | 5.556 |
| αV4.2 | 6 | 36 | 0 | 0 | 0 | 0.000 |
| αV5.1 | 7 | 42 | 0 | 0 | 0 | 0.000 |
| αV7.2 | 6 | 36 | 0 | 2 | 2 | 5.556 |
| αV10.1 | 7 | 84 | 0 | 3 | 3 | 3.571 |
| αV10.2 | 7 | 189 | 3 | 0 | 3 | 1.587 |
| Sum | 66 | 735 | 7 | 17 | 24 | |
| Mean Mutation Freq (%) | | | 0.95 | 2.31 | | 3.27 |
| Standard Deviation | | | 1.027 | 2.067 | 2.272 | |

**CDR Means**

| αV Group | S | N | ALL |
|---|---|---|---|
| αV1.1 | 4 | 8 | 12 |
| αV1.2 | 7 | 10 | 17 |
| αV1.4 | 0 | 3 | 3 |
| αV2.1 | 0 | 1 | 1 |
| αV3.1 | 0 | 3 | 3 |
| αV4.1 | 5 | 2 | 7 |
| αV4.2 | 1 | 1 | 2 |
| αV5.1 | 0 | 10 | 10 |
| αV7.2 | 1 | 3 | 4 |
| αV10.1 | 3 | 3 | 6 |
| αV10.2 | 6 | 10 | 16 |
| Sum | 27 | 54 | 81 |
| Mean Mutation Freq (%) | 1.17 | 2.35 | 3.52 |
| Standard Deviation | 2.659 | 3.754 | 5.626 |

**Table 3.** Number and frequency of DNA mutations that occur in tandem within framework regions (FR) and complementarity determining regions (CDR) in nurse shark alpha V (αV) groups.
All mutations include both tandem and point mutations within a region. [*Seqs:* sequences]

| | | FR | | | | | | CDR | | | | | |
|---|---|---|---|---|---|---|---|---|---|---|---|---|---|
| | | # Nucleotides Tandemly Mutated | | | | | | # Nucleotides Tandemly Mutated | | | | | |
| αV Group | # Seqs | 2 | 3 | 4 | Sum | All mutation | Frequency of tandem mutation | 2 | 3 | 4 | Sum | All mutation | Frequency of tandem mutation |
| αV1.1 | 3 | 1 | 0 | 1 | 6 | 24 | 25.0 | 3 | 0 | 0 | 6 | 12 | 50.0 |
| αV1.2 | 5 | 4 | 1 | 0 | 11 | 37 | 29.7 | 5 | 1 | 0 | 13 | 17 | 76.5 |
| αV1.4 | 3 | 3 | 0 | 0 | 6 | 14 | 42.9 | 0 | 0 | 0 | 0 | 3 | 0.0 |
| αV2 | 5 | 0 | 0 | 0 | 0 | 1 | 0.0 | 0 | 0 | 0 | 0 | 1 | 0.0 |
| αV3 | 4 | 2 | 0 | 0 | 4 | 5 | 80.0 | 1 | 0 | 0 | 2 | 3 | 66.7 |
| αV4.1 | 2 | 4 | 1 | 0 | 11 | 23 | 47.8 | 0 | 1 | 0 | 3 | 7 | 42.9 |
| αV4.2 | 2 | 1 | 1 | 0 | 5 | 11 | 45.5 | 0 | 1 | 0 | 3 | 2 | 150.0 |
| αV5 | 2 | 1 | 0 | 0 | 2 | 14 | 14.3 | 1 | 1 | 0 | 5 | 10 | 50.0 |
| αV7.2 | 2 | 4 | 1 | 0 | 11 | 22 | 50.0 | 1 | 0 | 0 | 2 | 4 | 50.0 |
| αV10.1 | 4 | 3 | 0 | 0 | 6 | 6 | 100.0 | 0 | 0 | 0 | 0 | 6 | 0.0 |
| αV10.2 | 9 | 4 | 1 | 0 | 11 | 35 | 31.4 | 2 | 1 | 0 | 7 | 16 | 43.8 |
| Total | 41 | 27 | 5 | 1 | 73 | 192 | 38.0 | 13 | 5 | 0 | 41 | 81 | 50.6 |

DOI: https://doi.org/10.7554/eLife.28477.011

CDR2 interacts primarily with non-polymorphic regions of MHC (*Buslepp et al., 2003*; *Garcia and Adams, 2005*), making mutation in CDR2 less favorable; further, FR regions are important for the structural stability of the domain and mutations to these regions may affect the ability of the Ig superfamily domain to fold properly (*Mantovani et al., 2002*; *Reinherz et al., 1999*).

## Hotspots

We found 1327 G/C base pairs within DGYW/WRCH hotspot motifs (4,402 G/C base pairs occurred outside these motifs) and 2931 A/T base pairs within WA/WT hotspot motifs (3690 A/T base pairs occurred outside these motifs; see *Table 4*). Mutations of G:C nucleotides were strongly associated with DGYW/WRCH hotspots (*Figure 7b,c*). Overall, G:C mutations occurred 4.3x as often within hotspots as those outside hotspots (p=0.0267). However, G:C mutations were 1.4x more likely within CDRs than within FRs (p<0.0001). A:T nucleotides did not appear to prefer WA/TW hotspots (p=0.2248), although they were 2.3x as likely to mutate within hotspots than outside hotspots. Further, while A:T mutations occurred more often than expected in both FRs and CDRs (p=0.0015), the frequency of A:T mutations were 1.9x more likely in CDRs than FRs. *Chen et al. (2012)* observed similar results in TcR γV sequences of sandbar shark. The authors suggested that A:T mutations still are likely the result of DNA polymerase η use during mismatch repair mechanisms due to the lack of A:T point mutations in that study (the majority occurring in tandem with other mutations) (*Rogozin et al., 2001*; *Chen et al., 2012*; *Wei et al., 2015*). In nurse shark Ig light chain genes, the majority of mutation to A:T nucleotides (55%) also occurred in tandem (*Alder et al., 2005*). However, 63% (76 of 121) of A:T mutations occurred as point mutations in our study, which is inconsistent with DNA polymerase η use during mismatch repair (*Rogozin et al., 2001*; *Chen et al., 2012*; *Wei et al., 2015*). This result suggests that an alternate A:T motif is targeted or that shark TcRs employ a different mechanism to alter A:T nucleotides.

## Base substitution indices

We identified 283 mutations within the eleven CDR3 groups. There was a bias toward mutations of G and C nucleotides (p=0.0022) and G:C changes comprised 57% of all mutations (*Table 5a*). More mutations to both G and C nucleotides occurred in FRs (p=0.0037) while only C nucleotides showed greater mutation than expected in CDRs (p=0.1136). There were fewer mutations of A and T

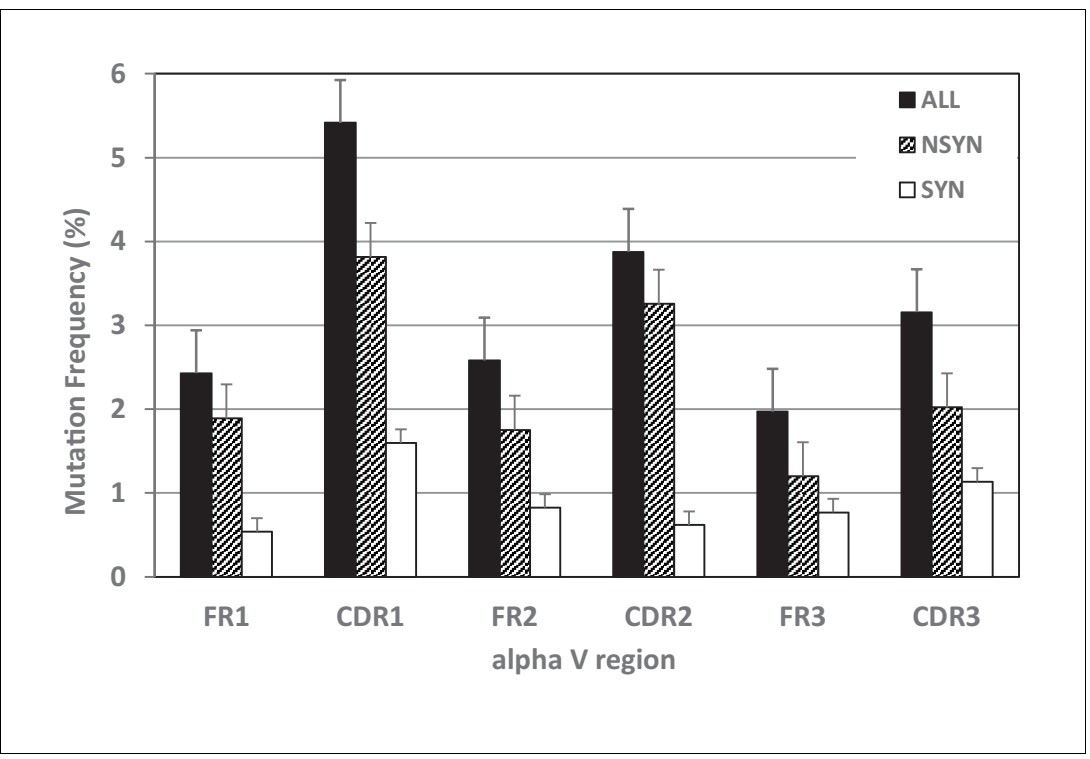

**Figure 6.** Mutation frequencies differed within TcR V regions. Mutability of complementarity determining regions (CDR), especially CDR1, exceeded that of framework regions (FR) for all mutations together (black bars) and for nonsynonymous mutations alone (NSYN, hatched bars). We found no statistical difference in synonymous mutations (SYN, white bars) between CDRs and FRs.

DOI: https://doi.org/10.7554/eLife.28477.012

The following source data is available for figure 6:

**Source data 1.** Frequency of mutation in TcR alpha V framework and complementarity determining regions for all mutation, nonsynonymous (NSYN) mutation only, or synonymous (SYN) mutation only.

DOI: https://doi.org/10.7554/eLife.28477.013

nucleotides in FRs (p=0.0082; *Table 5b*), but mutations of A and T nucleotides in CDRs did not differ from random (p=0.3620; *Table 5c*). The frequency of mutated nucleotides also varied by region: In FR1, more A and T nucleotides mutated while fewer C nucleotides mutated. Mutations of G and C nucleotides were lower in FR2 and FR3, respectively (*Table 5b,c*). In both CDR1 and CDR2, there were more A mutations and fewer G mutations than expected. However, more G nucleotides mutated in CDR3. We saw a bias toward G:A and C:T transitions (42.8%) among the TcR α mutations. Transition mutations appeared only slightly more often in CDRs (45.1%) than in FRs (41.7%). Transversions of C:A or G:T mutations occurred only 25.1% of the time (*Table 5*).

## Mutations, in situ hybridization, and AID expression in thymus

Since we cloned TcR αV sequences in both central (thymus) and peripheral (blood, spiral valve) lymphoid tissues, we analyzed mutation frequencies by tissue type. Although data were limited to only six sequences in two CDR3 groups (αV2, αV7.2), we found more mutations to FRs of peripheral sequences, though this was not significant (p=0.0541; *Table 6*). We found no differences between tissues within CDRs (p=0.2) from this limited data set. It is possible that positive and negative selection pressures remove more clonal sequences within the thymus, but with so few sequences to compare it is difficult to determine.

The mutated sequences in the primary T lymphoid tissue suggests the activity of activation-induced cytidine deaminase (AID) in the thymus. We confirmed the expression of AID in the thymus through real-time RT-qPCR, where thymus tissue expressed AID at more than half (0.7x) the levels found in spleen (positive control, where B cell SHM is known to occur) and nearly 6x the levels

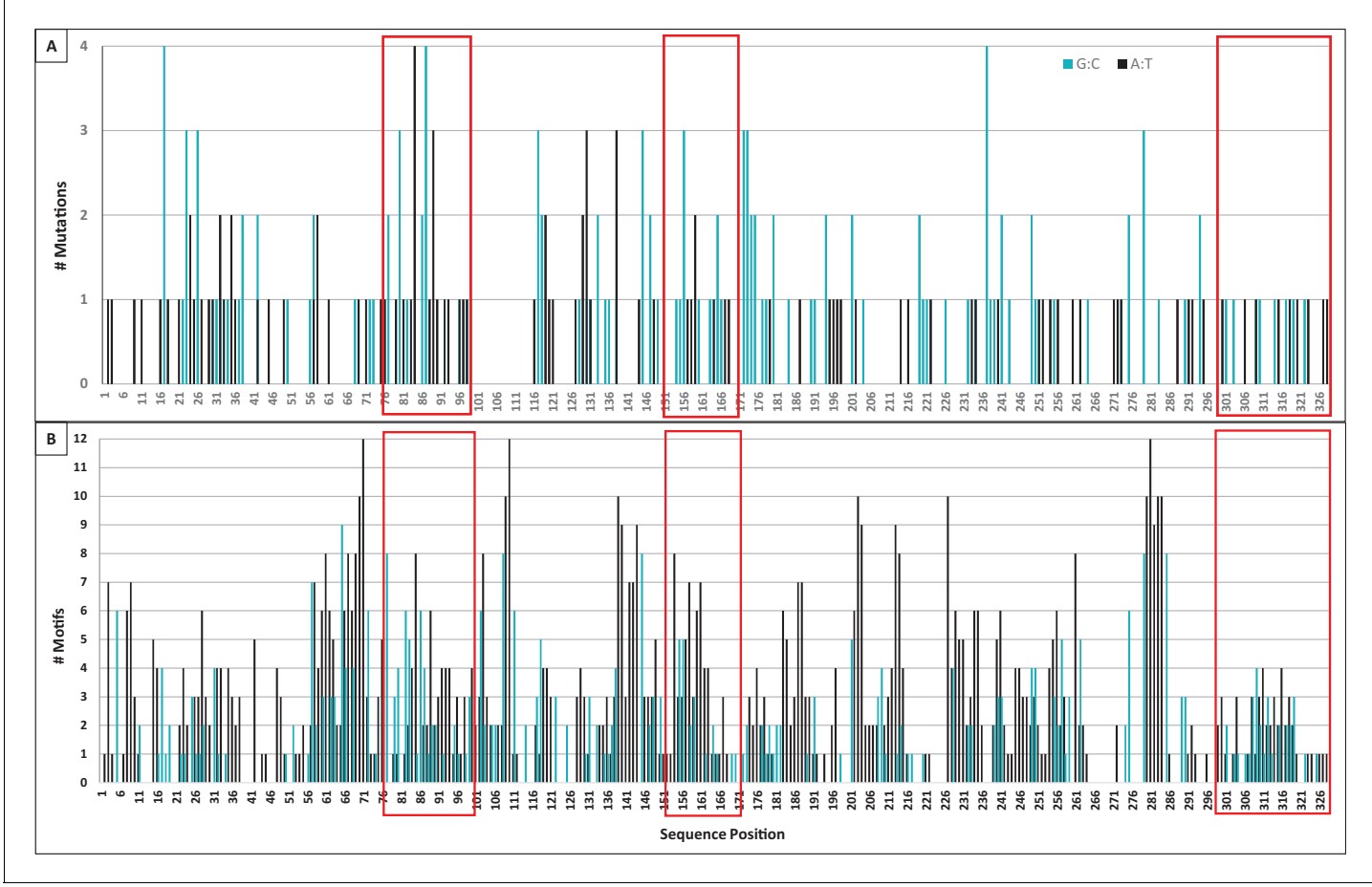

**Figure 7.** Mutation and motif locations within individual domains of TcR αV sequences. (A) Number of mutations (both single and tandem, 283 total) to A:T nucleotides (black bars, 122 mutations) or G:C nucleotides (blue bars, 161 mutations) observed at each position along the sequence length of TcR αV sequences. We counted mutations to Geneious-derived consensus sequences within framework and complementarity-determining region (CDR) domains (41 sequences within 11 αV groups). (B) Number of WA/TW motifs (black bars, indicating possible polymerase η action) or DGYW/WRCH motifs (blue bars, indicating hotspots for AID activity, and thus possible mutation) at each position along the sequence length. Position indicates the forward (3' to 5') location of the mutable base of each motif within a Geneious-derived consensus sequence for each TcR αV group (eleven groups). Red boxes indicate the location of CDR1 (positions 76–99), CDR2 (positions 151–171), and CDR3 (positions 300–328) within each panel. [WA/TW: A:T is the mutable position; DGYW/WRCH: G:C is the mutable position; W = A/T, D = A/G/T, Y = C/T, R = A/G, and H = T/C/A].
DOI: https://doi.org/10.7554/eLife.28477.014

observed in forebrain (negative control; *Figure 8a*). Further, colorimetric in situ hybridization (CISH) of nurse shark thymus revealed a diffuse signal of TcR β (*Figure 8b*, panels 1,2) and AID (*Figure 8b*, panels 3,4) mRNA expression throughout the thymic cortex. However, AID expression was greatest in the central cortex and cortico-medullary junction (CMJ), while TcR β expression was highest in the outer cortex. The thymic sequence data combined with the tissue localization of AID message strongly suggest that TcRα loci undergo AID-dependent SHM in the shark thymus.

We refined our CISH results with RNA fluorescence in situ hybridization (FISH) using probes against the TcR alpha constant region (αC) and exons 2 and 3 of *AID*. Since Stellaris RNA FISH uses a mixture of shorter probes (~20 bp in length) that hybridize along the length of the target RNA, the resulting signal is detectable only when tens of probes hybridize to the target. This makes the technology very specific – only those transcripts bound to numerous probes are visible – and limits the potential for false-positive or false-negative results (*Orjalo et al., 2011*; *Raj and Tyagi, 2010*). Our FISH results indicated a more specific TcR αC signal within the inner cortex and medulla adjacent to the CMJ (*Figure 9*; *Figure 9—figure supplement 1*), regions where, in mammals, developing cells actively rearrange their alpha chain genes and where mature αβ T cells are found. AID expression

**Table 4.** Target nucleotide mutation frequency in DGYW/WRCH or WA/TW mutation hotspots within framework regions (FR) and complementarity determining regions (CDR).

[DGYW/WRCH (G:C is the mutable position; D = A/G/T, Y = C/T, W = A/T, R = A/G, and H = T/C/A); WA/TW (A:T is the mutable position; W = A/T); 'ALL' refers to nucleotides within a hotspot motif; 'All other' refers to nucleotides outside a hotspot motif]

| Mutated base | Hotspot motif | Region | Total nucleotides | Observed # Mutations | Expected # Mutations | Mutation freq (%) | $\chi^2$ $p$ | T-test $P$ |
|---|---|---|---|---|---|---|---|---|
| G/C | DGYW/WRCH | FR | 927 | 56 | 18.16 | 6.04 | 0.0000 | 0.0267 |
| | | CDR | 400 | 35 | 26.22 | 8.75 | | |
| | | ALL | 1327 | 91 | 37.52 | 6.86 | 0.0000 | |
| | Outside Motif | | 4402 | 71 | 124.48 | 1.61 | | |
| A/T | WA/TW | FR | 2531 | 54 | 36.07 | 2.13 | 0.0015 | 0.2248 |
| | | CDR | 400 | 26 | 21.12 | 4.03 | | |
| | | ALL | 2931 | 80 | 55.97 | 2.52 | 0.0000 | |
| | Outside Motif | | 3690 | 41 | 65.03 | 1.11 | | |

*T-test analysis was used to compare mutations within hotspot motifs to those outside hotspot motifs. Mutations to G and C nucleotides occurred significantly more often within DGYW/WRCH motifs than outside these motifs, while mutations to A and T nucleotides showed no preference for WA/TW motifs.

†$\chi^2$ analysis was used to compare observed and expected numbers of mutations between FR and CDR regions and between mutations inside and outside hotspot motifs. More mutations to all nucleotides occurred within hotspots than outside hotspots, and significantly more mutations occurred to nucleotides within CDRs than FRs.

DOI: https://doi.org/10.7554/eLife.28477.015

occurred in 'rings' around areas of expressed TcR αC messages within the inner cortex and CMJ (where positive selection occurs in mammals) and the medulla (where negative selection occurs in mammals). Further, AID always co-localized with TcR αC (**Figure 9**). Thus, we observed a consistent pattern of expression where a 'ring' of cells expressing both TcR αC and AID surround a central cell expressing only TcR αC. The more specific signal generated by FISH may suggest that, once a T cell completes RAG-mediated somatic rearrangement of its alpha chain locus, it clonally expands to form a ring of daughter cells around it. These daughter T cells then express AID (and TcR αC), promoting somatic hypermutation within their TcR alpha sequences during times when cells also undergo positive and negative selection.

## Discussion

The role and diversifying mechanisms of SHM in B cells are well known (**Li et al., 2004**), as are the consequences of off-target AID activity (**Álvarez-Prado et al., 2018**). In B cells, AID mediates SHM within germinal centers of lymph nodes and spleen in mammals (**Crouch et al., 2007**), and we predict this is similarly occurring in the B cells zones identified in the shark splenic white pulp (**Rumfelt et al., 2002**). Somatic mutations occur in rearranged variable regions of B cells responding to antigen at rates of $10^{-3}$ mutations per base pair per cell division (**Odegard and Schatz, 2006**). These changes are dominated by point mutations (and in shark, tandem mutations), biased toward transitions (G:A and C:T), and preferentially targeted to the AID motifs DGYW/WRCH (and less to WA/TW) (**Li et al., 2004**; **Malecek et al., 2005**; **Odegard and Schatz, 2006**; **Rogozin and Diaz, 2004**). The sequences of B cell V genes have evolved to maximize mutational effects, targeting the accumulation of replacement mutation within the antigen-binding CDRs and limiting mutation within the more structural FRs. In humans, this focused mutation correlates with the long-term survival of B cell receptor repertoires (**Saini and Hershberg, 2015**). The ability of B cells to use SHM for receptor diversification and improved antigen affinity is the basis of adaptive immunity (**Saini and Hershberg, 2015**). Despite having similar developmental machinery as B cells (**Gellert, 2002**), the assumption has long been held that αβ T cells do not undergo SHM because mutation could have deleterious effects on the binding of TcRs to MHC:Ag complexes (MHC:Ag) (**Wagner et al., 1995**; **Mantovani et al., 2002**). Despite some suggestion that SHM was occurring in T cells, studies designed to either quantify or characterize mutation in mouse or human TcRs did not gain traction

**Table 5.** Bias in base substitution during somatic hypermutation of TcR alpha V genes within all sequence regions (ALL), framework regions (FR), or complementarity determining regions (CDR).

Probability of occurrence is the proportion of that base out of the total nucleotides. [Nuc: nucleotides; OBS: Observed; EXP: expected; MI: mutability index; ChiSq: Chi squared]

| a | ALL | Base | Occurrence | Probability of occurrence | OBS | EXP | MI* | ChiSq |
|---|-----|------|-----------|---------------------------|-----|-----|-----|-------|
| | | G | 2895 | 0.230 | 77 | 65.05 | 1.18 | 0.0022 |
| | | C | 2834 | 0.225 | 85 | 63.68 | 1.33 | |
| | | A | 3498 | 0.278 | 64 | 78.60 | 0.81 | 0.0068 |
| | | T | 3368 | 0.267 | 57 | 75.68 | 0.75 | |
| | | Total | 12595 | 1.00 | 283 | 283 | | |

GC Mutation: 57.0%; Transitions: 42.8%; Transversions: 25.1%

| b | FR | Base | Occurrence | Probability of Occurrence | OBS | EXP | MI* | ChiSq |
|---|-----|------|-----------|---------------------------|-----|-----|-----|-------|
| | | G | 2378 | 0.23 | 58 | 44.50 | 1.30 | 0.0037 |
| | | C | 2268 | 0.22 | 56 | 42.44 | 1.32 | |
| | | A | 2779 | 0.27 | 38 | 52.00 | 0.73 | 0.0082 |
| | | T | 2835 | 0.28 | 40 | 53.05 | 0.75 | |
| | | Total | 10260 | 1.00 | 192 | 192 | | |

GC Mutation: 59.0%; Transitions: 41.7%; Transversions: 24.0%

| c | CDR | Base | Occurrence | Probability of Occurrence | OBS | EXP | MI | ChiSq |
|---|-----|------|-----------|---------------------------|-----|-----|-----|-------|
| | | G | 517 | 0.22 | 19 | 20.15 | 0.94 | 0.1336 |
| | | C | 566 | 0.24 | 29 | 22.06 | 1.31 | |
| | | A | 719 | 0.31 | 26 | 28.02 | 0.93 | 0.3620 |
| | | T | 533 | 0.23 | 17 | 20.77 | 0.82 | |
| | | Total | 2335 | 1.00 | 91 | 91 | | |

GC Mutation: 52.9%; Transitions: 45.1%; Transversions: 27.5%

*Mutability Index, as first defined in **Chen et al., 2012**. $\chi^2$ analysis was used to compare observed and expected numbers of mutations. G and C mutated significantly more often than expected, while A and T mutated significantly less often than expected. Base composition: 23.0% G, 22.5% C 27.8% A, 26.7% T.

DOI: https://doi.org/10.7554/eLife.28477.016

---

**Table 6.** Frequencies of somatic hypermutation in nurse shark thymus and peripheral lymphoid tissue (blood and spiral valve).

Mutations were analyzed only in alpha V groups containing the same third complementarity-determining region (CDR). Mutation frequency was measured as the total number of nucleotide changes to a Geneious-derived consensus sequence divided by the total number of nucleotides in all sequences. Nonsynonymous (N) and synonymous (S) mutations (mut) were counted separately for each framework (FR) and CDR for two predicted alpha V genes. [FR1, FR2, FR3, CDR1, CDR2, and CDR3 refer to the first, second, or third FR or CDR region, respectively.]

| Tissue type | Mut type | FR Mutations (#) | | | | CDR Mutations (#) | | | | FR mutation frequency | | | CDR mutation frequency | | |
|---|---|---|---|---|---|---|---|---|---|---|---|---|---|---|---|
| | | FR1 | FR2 | FR3 | All FR | CDR1 | CDR2 | CDR3 | All CDR | FR1 | FR2 | FR3 | CDR1 | CDR2 | CDR3 |
| Thymus (6 sequences) | N | 8 | 8 | 7 | 23 | 1 | 3 | 3 | 7 | 0.570 | 0.871 | 0.317 | 0.071 | 0.327 | 0.136 |
| | S | 8 | 2 | 6 | 16 | 1 | 0 | 0 | 1 | 0.570 | 0.218 | 0.272 | 0.071 | 0.000 | 0.000 |
| | ALL | 16 | 10 | 13 | 39 | 2 | 3 | 3 | 8 | 1.140 | 1.089 | 0.590 | 0.529 | 1.075 | 1.235 |
| Total Nucleotides | | 1404 | 918 | 2205 | 4527 | 378 | 279 | 243 | 900 | | | | | | |
| Periphery (2 sequences) | N | 7 | 4 | 6 | 17 | 0 | 2 | 2 | 4 | 1.496 | 1.307 | 0.833 | 0.000 | 1.852 | 1.587 |
| | S | 4 | 0 | 1 | 5 | 0 | 0 | 0 | 0 | 0.855 | 0.000 | 0.139 | 0.000 | 0.000 | 0.000 |
| | ALL | 11 | 4 | 7 | 22 | 0 | 2 | 2 | 4 | 2.350 | 1.307 | 0.972 | 0.000 | 1.852 | 1.587 |
| Total Nucleotides | | 468 | 306 | 720 | 1494 | 126 | 108 | 126 | 360 | | | | | | |

DOI: https://doi.org/10.7554/eLife.28477.017

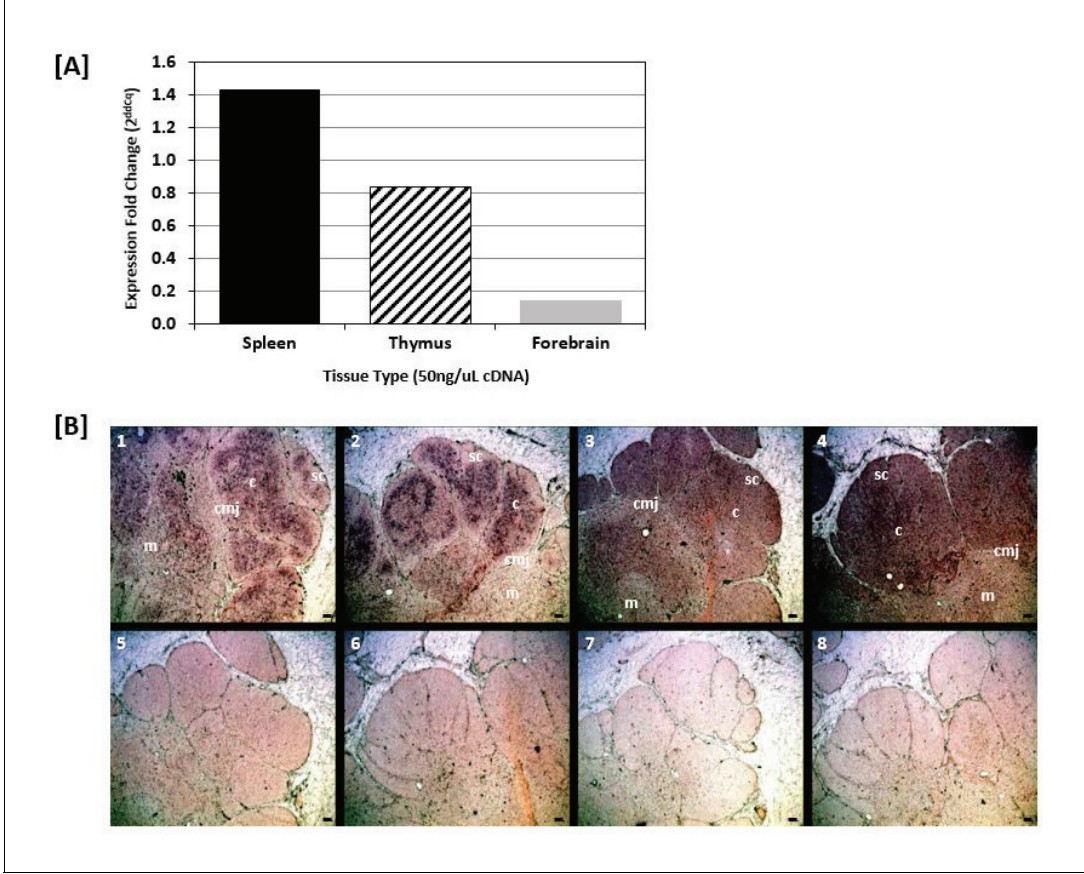

**Figure 8.** Shark thymus expresses AID. (**A**) Expression of AID in shark spleen (black bar), thymus (hashed bar), and forebrain (gray bar) using real-time quantitative PCR. Expression was measured using the delta delta Cq method and are normalized against shark muscle tissue using β2M as a reference gene. Data represent expression fold-change differences at four cDNA concentrations. (**B**) In situ hybridization of mRNA using ribo-probes on adult shark thymus sections. **1,2**: Two different fields probing for TcRβ antisense. **3,4**: AID antisense. **5,6**: TcRβ sense. **7,8**: AID sense. All micrographs at 10X magnification; black scale bar in lower right of each panel is 100 uM. Anatomical structures are designated on the top panels [c: cortex; m: medulla; sc: subcapsular region; cmj: corticomedullary junction].

DOI: https://doi.org/10.7554/eLife.28477.018

and the textbook definition of SHM still defines it as an exclusively B cell mechanism (*Murphy and Weaver, 2017*).

Recent studies reported the incidence of SHM in the γ chain of γδ T cells of sandbar shark (*Chen et al., 2012*; *Chen et al., 2009*) and in both γ and δ chains of dromedary camel (*Antonacci et al., 2011*; *Ciccarese et al., 2014*; *Vaccarelli et al., 2012*). In each study, SHM mirrored the mutational patterns observed in B cells during affinity maturation. However, in both sandbar shark γ and dromedary camel γ and δ chains, the authors hypothesized that T cells employ mutation as a means to generate a more diverse receptor repertoire rather than to improve receptor affinity to Ag (*Chen et al., 2012*; *Antonacci et al., 2011*; *Vaccarelli et al., 2012*). In contrast to αβ T cells, γδ T cells that interact with non-classical MHC often recombine tissue-specific, restricted sets of genes that have limited junctional diversity (*Allison et al., 2001*; *Adams et al., 2005*). Thus, it is reasonable to consider that SHM could be used as a receptor-diversifying mechanism to fine-tune ligand recognition within a prospective tissue or to allow changes within the loci that allow receptors to evolve more rapidly to changing ligand environments (*Adams et al., 2005*; *Kazen and Adams, 2011*). Further, many γδ T cells typically bind Ag in a manner more similar to that of Ig than to αβ T cells, recognizing and directly binding to small molecules and intact proteins without presentation by classical MHC:Ag complexes (*Adams et al., 2005*; *Allison and Garboczi, 2002*; *Allison et al., 2001*). Inflammation stimulates activation of γδ T cells earlier in an immune response, releasing pro-inflammatory cytokines and killing infected macrophages. Thus, γδ T cells combine an innate-like

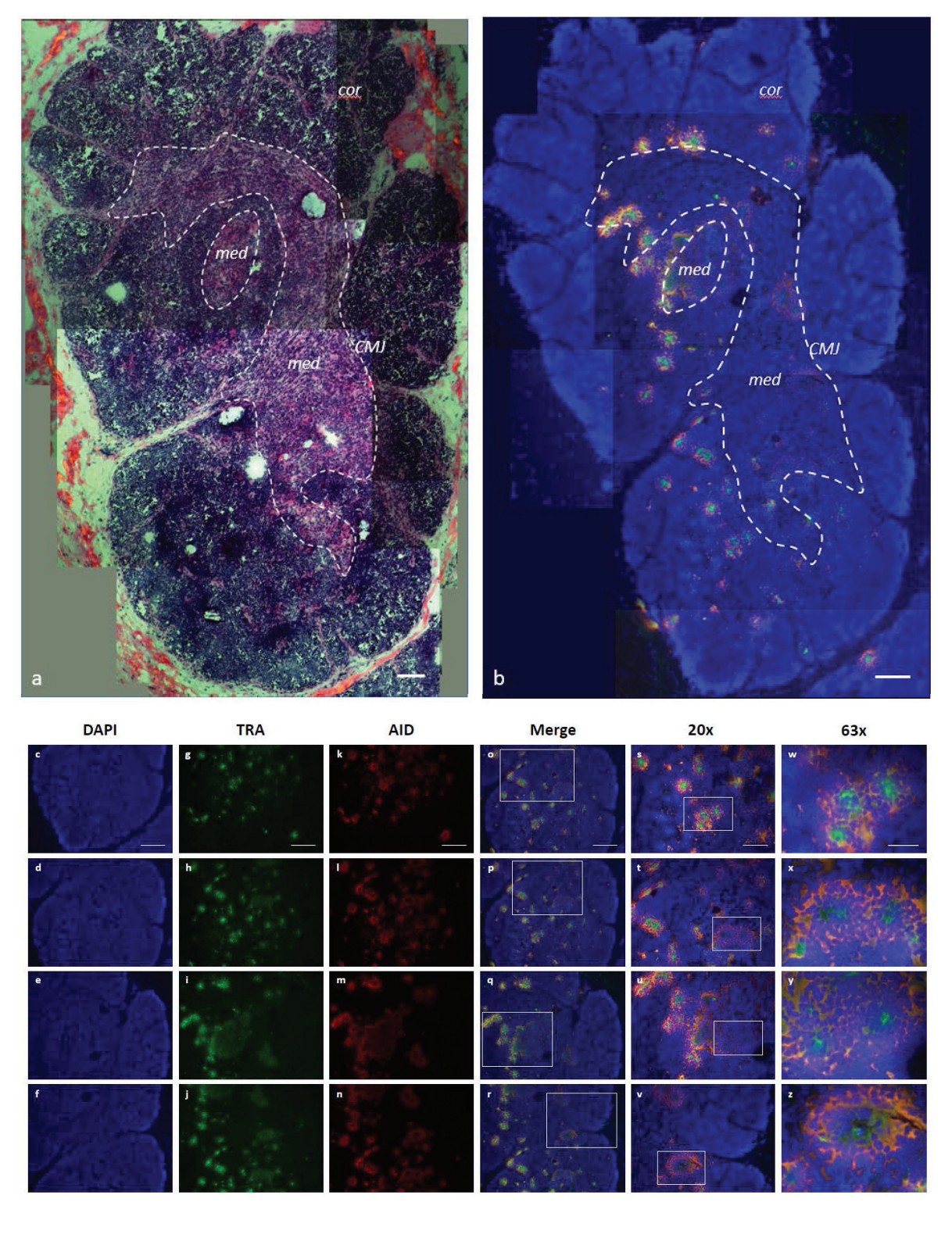

**Figure 9.** AID expression localized to inner cortex and cortico-medullary junction. (a) H and E staining of fixed shark thymus tissue illustrating thymic architecture (10x). The densely packed cells at the margins of the image comprise the cortex (cor), while the less densely packed cells in the center constitute the medulla (med). The region at the junction between cortex and medulla incorporates the corticomedullary junction (CMJ), delineated generally by a hashed white circle. [b-z] Single molecule RNA fluorescence in situ hybridization (FISH) probing fixed thymus sections simultaneously for

*Figure 9 continued on next page*

*Figure 9 continued*

AID (probes labeled with Quasar 670; pseudo colored *red*) and TcRα (probes labeled with CalFluor Red 610; pseudo colored *green*) and counterstained with DAPI (*blue*). (b) Composite of seven Z-stacked images (10x) depicting overall thymic architecture and the localization of AID expression to the inner cortex and cortico-medullary junction regions of shark thymus. We superimposed (and minimally adjusted) the outlined CMJ boundaries from (a) onto (b) to elucidate the junction between cortex and medulla. [c–z] We obtained images of each fluorophore using 10x, 20x, and 63x magnification and merged Z-stacked images together. Individual (10x) fluorophore images of DAPI [c-f], TcRα [g-j], and AID [k-n] and Z-stacked merged images [o-r] illustrate AID and TcRα expression in four locations of shark thymus. White boxes indicate the magnified regions of the 10x and 20x images shown in the 20x [s-v] and 63x [w-z] images, respectively. Scale bars [a,b,c,g,k,o] 150 μm, [s] 75 μm, and [w] 30 μm. [*cor*: cortex; *med*: medulla; *CMJ*: corticomedullary junction].

DOI: https://doi.org/10.7554/eLife.28477.019

The following figure supplements are available for figure 9:

**Figure supplement 1.** Localization of AID and TcRα probes is independent.
DOI: https://doi.org/10.7554/eLife.28477.020

**Figure supplement 2.** Lack of AID and TcRα probe hybridization in shark brain.
DOI: https://doi.org/10.7554/eLife.28477.021

response with an adaptive recognition strategy, providing both an immediate response to pathogen invasion and an ongoing, adaptive response to inflammation (*Adams et al., 2005*; *Allison and Garboczi, 2002*; *Allison et al., 2001*). It is evident that how SHM presents a useful solution for accomplishing these tasks by creating a more diverse repertoire of these antibody-like γδ TcRs. Taken together, these studies clearly demonstrate that we can no longer regard SHM as a uniquely B cell mechanism. Considering the diversity of TcRs and TcR diversification mechanisms being found even in mammals (*Hansen and Miller, 2015*; *Miller, 2010*), perhaps we should prepare for more surprises in TCR antigen recognition.

In the present study, we verified SHM occurring within the γ and δ chains of γδ T cells in both thymic and peripheral immune tissue of nurse shark. Remarkably, we also detected SHM occurring in the α chain of αβ T cells. We observed mutational characteristics within α chain of nurse sharks similar to those found in B cell SHM. We observed an overall mutation frequency of 0.0226 substitutions per nucleotide (S/N) and a bias toward transition mutations. Further, we detected both single and tandem mutations, a pattern unique to sharks that also occurs in shark B cells. Changes to G and C nucleotides comprise 66.1% of all mutations. Mutation was twice as frequent in CDRs as in FRs (0.0352 versus 0.0188 S/N, respectively), and substitutions in CDRs were significantly more likely to result in amino acid changes. Further, mutations were strongly associated with AID hotspots, and substitutions to G and C nucleotides occurred nearly 1.4x as often within CDR hotspots than FR hotspots. Out of curiosity, we compared counts of AID hotspot motifs within CDR and FR regions between our 11 nurse shark TcR αV consensus sequences and 6 human TcR αV segments (V1.1, V1.2, V2, V3, V4 and V5). We found that shark TcR V segments exhibit far more WR<u>C</u>H/D<u>G</u>YW motifs per sequence than do human V segments (p=0.02). Further, motifs in CDRs of shark occurred 2-3x as often as in humans [human: average of 2.27 motifs per FR (range 1.97–2.65), 2.28 per CDR (range 1.97–2.59); shark: average of 3.25 motifs per FR (range 2.85–3.88), 5.09 per CDR (range 4.02–6.16); data not shown]. The bias we found for nonsynonymous and non-conservative mutations in TcRα CDRs in the shark thymus are consistent with more than simple repertoire diversification; it suggests selection for changes in paratope.

We identified SHM from identical cDNA clones originating from both thymus and spiral valve tissues (see *Table 6*), suggesting that T cells with SHM-modified receptors must have originated within the thymus and then traveled to peripheral gut-associated lymphoid tissue. Unsurprisingly, we detected the most AID expression within the inner cortex, medulla, and CMJ of shark thymus, where rearrangement and testing of TcRα takes place in mammals. Positive selection on self-MHC/self peptide for mature thymocytes begins with the CD4/CD8 double positive (DP) stage of development while differentiation into CD4/CD8 single positive (SP) cells requires that the TcR interact with MHC (*Huesmann et al., 1991*). If there is no TcR: MHC/peptide match found, T cell differentiation stalls with failure to be positively selected (*Reinherz et al., 1999*). However, the unusual nature of the TcRα locus, with up to 100 J segments depending on species, permits multiple successive rearrangements within a single cell, rescuing non-productive or self-selectable receptors with further gene rearrangements, a process called *receptor editing* (*Bedel et al., 2012*; *Borgulya et al., 1992*;

*Guo et al., 2002*; *Petrie et al., 1993*). In mice, unlike the situation in developing B cells, receptor editing does not seem to rescue T cells from negative selection (*Kreslavsky et al., 2013*) and thus provides several opportunities for positive selection of DP thymocytes. Thymic nurse cells may help optimize these opportunities for selection by providing microenvironments favorable to secondary alpha chain rearrangement (*Nakagawa et al., 2012*).

In developing shark thymocytes, SHM in TcRα loci in conjunction with receptor editing (note that sharks, like all other gnathostomes, have a large number of TcRα J segments) could be involved in salvaging cells for positive selection or rescuing cells from death by negative selection. If AID-induced SHM occurs in conjunction with receptor editing and positive selection, AID should be upregulated in cells undergoing RAG-mediated alpha rearrangement (and thus in cells also expressing RAG). However, if SHM occurs *after* rearrangement of TcR α and thus used for rescuing cells during negative selection, the same T cell would not express both AID and RAG. While we cannot determine conclusively without RAG expression data, the patterns of AID and TcRα expression (*Figure 9*) suggest that AID is upregulated after cells proliferate and diversify following alpha rearrangement (within the 'ring' of cells). Thus, it is likely that AID is used primarily to rescue cells from negative selection, providing a 'mini-expanded self-referential repertoire' (*Figure 10*) and reducing the 'profligate waste of thymocytes' (*Murphy and Weaver, 2017*). However, based on the works above by Kreslavsky et al. and Nakagawa et al. in mice, we cannot discount the possibility that TcRs use SHM in conjunction with receptor editing for positive selection since developing shark T cells could still undergo negative selection after SHM (*Figure 10*). We cannot completely rule out AID use in mature shark T cells, although our sequence data show no greater mutation frequency in the periphery, and abatement of AID expression in the thymic medulla are consistent with AID being a mechanism used only in T cell development. Further studies examining expression data from single cells could elucidate the timing of AID-catalyzed SHM in relation to T cell development.

These results are not without precedent. *Qin et al. (2011)* reported endogenous AID expression by peripheral CD4[+] T cells and immature B cells in mice. T cells that expressed AID also produce a distinctive cytokine profile, are associated with cell activation, and increase in abundance with age, suggesting these cells have distinctive long-term functions in aging cells. In immature B cells, AID

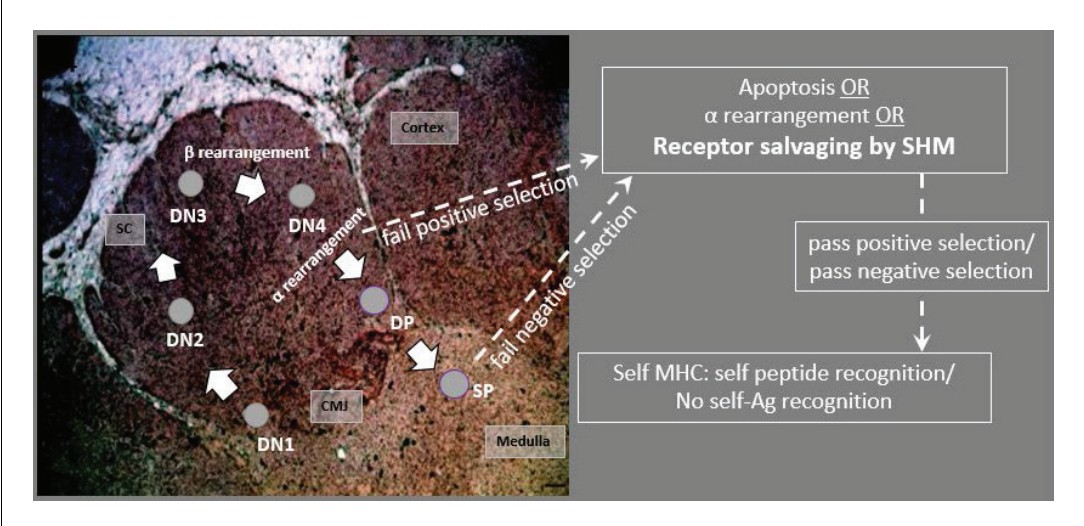

**Figure 10.** Model predicting how AID acts on T cells in the thymus. CD4/CD8 double negative (DN) thymocytes in the subcapsular region (SC) and cortex rearrange the β chain, using a surrogate pTα receptor to test for expression signaling. Cells with productive β arrangements then proliferate and express both CD4 and CD8, becoming double positive thymocytes (DPs). As DPs move toward the inner cortex and cortico-medullary junction (CMJ) where α chain rearranges, cells may begin to express AID. Non-productive rearrangements can be rescued from apoptosis by receptor editing or by receptor salvaging, in which AID catalyzes SHM to produce cells with improved affinity to MHC:Ag complexes (to pass positive selection). Salvaged thymocytes then proliferate and express either CD4 or CD8 on their surface as single-positive (SP) cells. AID-mediated receptor salvaging may also reduce recognition of self-peptide, rescuing self-reactive thymocytes from apoptosis (to pass negative selection). [sc: subcapsular region; cmj: corticomedullary junction; green shading indicates region of AID expression].
DOI: https://doi.org/10.7554/eLife.28477.022

expression may help negatively select autoreactive B cells and contribute to primary repertoire diversity. Together these results may indicate that AID expression in mice is a relic of a more extended expression in earlier species (*Qin et al., 2011*). Some mammals (e.g. rabbit, sheep, and cattle) use SHM in B cell primary repertoire diversification in the gut associated lymphoid tissues (*Alitheen et al., 2010*; *Archer et al., 1963*; *Becker and Knight, 1990*; *Butler et al., 2011*; *Lanning et al., 2000*; *Reynaud et al., 1995*; *Reynaud et al., 1991*; *Reynolds and Morris, 1983*). Even more recently, research implicated AID in central B cell tolerance in mammalian pre-B cells (*Cantaert et al., 2015*; *Kuraoka et al., 2011*), and there are examples of AID-mediated SHM being used alongside RAG-mediated V(D)J recombination in primary B cell repertoire generation in many mammals and in AID-mediated primary B cell diversification during gene conversion in the bird bursa of Fabricius (*Reynaud et al., 1987*; *Thompson and Neiman, 1987*). We suggest that these B-cell-specific features are much later mechanisms of AID-driven primary lymphocyte repertoire diversification and honing, perhaps mechanisms convergent with the thymic process we describe here. However, we predict that studies of this type in other vertebrate species will reveal the use of SHM as a T cell diversifying mechanism in a much broader collection of species.

Although recent studies report B cells present in the thymus of mice (*Perera and Huang, 2015*; *Perera et al., 2013*), Miracle et al. examined B cell expression levels (IgM and IgX) in various tissues (including thymus) at different ages in the clearnose skate. While skate thymus expressed both IgM and IgX in early life stages, adult skates no longer express Ig in the thymus (*Miracle et al., 2001*). *Criscitiello and Flajnik, 2007* also found little evidence that Ig light chain is expressed in adult nurse shark thymus (see Fig S3 of paper), corroborating older northern blot studies with heavy chain in the species (*Rumfelt et al., 2001*; *Rumfelt et al., 2004*). A concurrent study within our lab used RT-qPCR to analyze IgM expression within shark thymus and found that, while young sharks do express IgM in the thymus, adult nurse sharks (which we used in the current study) do not express IgM (data not shown). Further, preliminary results using FISH probes to IgM constant region also indicate that adult nurse sharks do not express IgM in the thymus. What little Ig expression we found in the thymus did not co-localize with AID expression.

Shark TcRs are capable of a wide variety of diversification mechanisms. In addition to RAG-mediated combinatorial and junctional diversity from complex loci, TcRδ (and possibly α) rearrange Ig-TcR chimera using IgM and IgW V exons with TcR D-J-C (*Criscitiello et al., 2010*). The TcR δ locus also encodes the doubly rearranging NAR-TcR, a δ chain with two diverse V domains (*Criscitiello et al., 2006*). Interestingly, our existing data do not suggest that SHM targets the IgHV or the TcR δV of the NAR-TcR δV chains for diversification. This suggests great control over the (nearly synchronous if not concomitant) expression of the potentially genotoxic AID and RAG. Thus, SHM at the γ, δ, and α loci adds to the battery of extraordinary diversification mechanisms used by shark lymphocytes in antigen recognition, although we do not yet understand the full effects of SHM on the animal's immunity to infectious disease. As for the dangers of any aberrant mutational activity genome-wide, sharks could be more resilient than other taxa due to their inherent slow rate of mutation (*Martin, 1999*), possibly linked to the exceptional longevity of some individuals (*Nielsen et al., 2016*).

In the broader scope of lymphocyte evolution, we must consider whether the ancestral vertebrate lymphocyte employed APOBEC-family-mediated diversification before the 'big bang' of RAG (reviewed in [*Hirano, 2015*]). Lamprey lymphocytes express at least two lymphocyte-specific cytidine deaminases (CDA1/CDA2) in the AID/APOBEC family. These deaminases emerged phylogenetically as the closest sister group in the AID/APOBEC family to the AID used by gnathostomes for Ig class-switch recombination, somatic hypermutation, and Ig gene conversion (*Rogozin et al., 2007*). CDA-mediated gene rearrangement in lampreys occurs in a manner similar to AID-induced immunoglobulin gene conversion in some birds and mammals (*Rogozin et al., 2007*; *Zheng et al., 1994*). One study suggested that VLRA (the analog to αβ TcR in jawed vertebrates) also might use CDA to affinity mature its receptors, indicating that CDA contributes both to repertoire generation and to somatic mutation after antigen exposure (*Deng et al., 2010*; *Flajnik, 2014*). If this is true, it may be possible that the ancestor to modern vertebrates also used an AID-like enzyme to assist with lymphocyte receptor development in a thymus-like organ. The expression of AID in the thymus of primitive sharks may be a remnant of this ancestral process, a mechanism lost in later vertebrates because of its potential for breaking down self-tolerance in mature lymphocytes. Perhaps, agnathans evolved specific APOBEC molecules for diversification of their B and T like VLRs, while gnathostomes evolved

AID for T cell primary repertoire diversification (Neils Jerne's 'mutant breeding organ' [*Jerne, 1971*]) and B cell affinity maturation, eventually co-opting AID for use in class switch recombination at IGH translocons, and later still, gene conversion and SHM for primary B cell repertoires.

From this trend of comparative TcR studies, we conclude with two hypotheses that we will test with further immunogenetic and functional studies in shark and other vertebrate models. First, the division between B and T cell repertoire diversification components and mechanisms was not as clear-cut in ancestral lymphocytes as in modern humans and mice. The second is that different vertebrate groups have not only evolved myriad diversifications for Ig repertoires and function, but TcR biology may be just as varied. This premise is already accumulating ample supporting evidence as IgHV domains (*Parra et al., 2010*; *Parra et al., 2012*), high allelic polymorphism (*Criscitiello et al., 2004a*; *Criscitiello et al., 2004b*), germline joined V exons (*Wang and Miller, 2012*; *Wang et al., 2011*), and now mechanisms such as SHM, all once considered the immune privilege of Ig or MHC genes, are also employed for TcRs.

## Materials and methods

### Study animals

TCR sequence data used in this study came from two adult female nurse sharks ('Joanie' and 'Mary Junior'; *Ginglymostoma cirratum*) delivered by caesarian section off the Florida Keys and matured in the aquatic vivarium of the University of Maryland's Center of Marine Biotechnology. We used published T cell αV sequences (*Criscitiello et al., 2010*) as a baseline for αV locus numbering (sharks Yellow and 1299), though we did not analyze any of these sequences for mutation.

### Total RNA isolation and cDNA synthesis

We harvested tissues from animals after MS-222 (Argent, Redmond, WA) overdose, and immediately purified RNA with TRIzol reagent (Life Technologies, Carlsbad, CA). Nurse shark thymi are located dorsomedial to the gills (*Luer et al., 1995*) in the crevasse between the epaxial and brachial constrictor muscle groups. We used 5 ug total RNA from spiral valve, spleen, thymus, and peripheral blood leukocytes (PBL) for oligo-dT primed cDNA generation with Superscript III First Strand Synthesis System (Thermo Fisher Scientific, Inc., Waltham, MA). (*Criscitiello et al., 2010*) We estimated cDNA concentration using a Nanodrop 2000 Spectrophotometer (Thermo Fisher Scientific, Inc.).

### RACE PCR, Cloning, and Sanger Sequencing

We generated a 5' RACE (Rapid Amplification of cDNA Ends) library using the GeneRacer Kit (Life Technologies) and reverse primers designed to the end of the shark TcRβ, TcRγ, or TcRδ variable (V) region or to the middle of the shark TcRα constant (C) region. We amplified RACE products using Phusion High-Fidelity DNA polymerase (New England Bio Labs, Inc., Ipswich, MA) to minimize PCR errors under these specific PCR conditions: primary denaturing at 94°C for 2 min; 30 cycles at 94°C for 30 s and 78°C for 1 min; and a final extension at 72°C for 10 min. Using this RACE library, we then amplified a specific α V region using a gene-specific primer to its leader region and the following PCR conditions: 98°C for 1 min; 25 cycles of 98°C for 5 s, 49–60°C for 30 s, 72°C for 150 s; 72°C for 10 min. Annealing temperatures varied for each amplified αV (see *Table 7*). We visualized PCR products with agarose gel (8%) electrophoresis and then excised bands of correct size. We then isolated amplified bands from agarose gels using the PureLink Quick Gel Extraction Kit (Life Technologies) or RICO chips (TaKaRa Bio USA, Mountain View, CA).

We transformed PCR amplicons into One-Shot Top10-competent cells (Thermo Fisher) using a pCR4-TOPO TA blunt end vector and cloning kit (Thermo Fisher) followed by a Zyppy plasmid miniprep kit (Zymo Research, Irvine, CA) for plasmid purification of individual clones (*Criscitiello et al., 2012*). We checked insert size using an *Eco* RI restriction enzyme (Promega Corp, Madison, WI), then amplified and purified the sequencing reaction using BigDye xTerminator Sequencing and Purification Kit (Thermo Fisher Scientific, Inc.). We submitted samples for sequencing to the DNA Technologies Core Lab on the Texas A and M University campus (College Station, TX). We deposited sequences in GenBank with the following accession numbers: *Alpha* KY189332-KY189354 and KY366469-KY355487; *Beta* KY351708-KY366487; *Gamma* KY351639-KY351707; *Delta* KY346705-KY346816.

**Table 7.** List of forward (F) and reverse (R) primers used to generate T cell receptor (TcR) sequences and expression data.
[AID: Activation induced cytidine deaminase; B2M: beta-2 microglobulin; α: alpha; β: beta; γ: gamma; δ: delta; V: variable region; C: constant region].

| Primer | F/R | ID | Location | Nucleotide sequence (5′ to 3′) | Amino acid | Tm |
|---|---|---|---|---|---|---|
| TcR αV1 | F | MFC370 | leader region of αV1 | ATG TTG CCT GAA GCT C | MLPEA | 55 |
|  | R | MFC191 | alpha C region | CAT TGG TGG ATA GCA AGC CCT TCG AT | SKGLLSTN | 76 |
| TcR αV4 | F | MFC122 | beginning of αV4 | GTC TCC TCA GTT GTT CGT AC | VSSVVR | 58 |
|  | R | MFC123 | end of αV4 | CAG TAA TAC ACA GCA GCG TC | DAAVYY | 58 |
|  | F | MFC374 | leader region of αV4 | TGG ATT GTG TGG GCA GTA | WIVWAV | 54 |
|  | R | MFC191 | alpha C region | CAT TGG TGG ATA GCA AGC CCT TCG AT | SKGLLSTN | 76 |
| TcR αV5 | F | MFC124 | beginning of αV5 | CTC AGG AAG GAG AGA TTA TCA C | QEGEII | 60 |
|  | R | MFC125 | end of αV5 | CAA TGA TAC ACG GCG GAG TC | DSAVYH | 60 |
|  | F | MFC124 | beginning of αV5 | CTC AGG AAG GAG AGA TTA TCA C | QEGEII | 60 |
|  | R | MFC191 | alpha C region | CAT TGG TGG ATA GCA AGC CCT TCG AT | SKGLLSTN | 76 |
| TcR αV7 | F | MFC376 | end of leader αV7 | AGC GAT GGA GTT TCT GTG ATT | SDGVSVI | 58 |
|  | R | MFC191 | alpha C region | CAT TGG TGG ATA GCA AGC CCT TCG AT | SKGLLSTN | 76 |
| TcR αV10 | F | MFC378 | leader region of αV10 | CTA TTT CTT CAC TAC CGC AG | YFFTTA | 56 |
|  | R | MFC191 | alpha C region | CAT TGG TGG ATA GCA AGC CCT TCG AT | SKGLLSTN | 76 |
| TcR α 5′ | F | GeneRacer 5′ Nested | homologous to RNA oligo | GGA CAC TGA CAT GGA CTG AAG GAG TA | – | 78 |
| TcR α 3′ | R | MFC191 | alpha C region | CAT TGG TGG ATA GCA AGC CCT TCG AT | SKGLLSTN | 76 |
| TcR βV1 | F | MFC126 | beginning of bV1 | CTC CGT ACA TCG TCT CTA TTG | PYIVSI | 60 |
|  | R | MFC127 | end of βV1 | CAC GCA CAG AAA TAG ACA GC | AVYFCA | 58 |
| TcR βV2 | F | MFC128 | beginning of βV2 | CTA CGT GGA GCA GTC TCC ATC | YVEQSP | 63 |
|  | R | MFC129 | end of βV2 | GCA CGC ACA ATA ATA GAC AGC C | AVYYCAC | 62 |
| TcR βV3 | F | MFC130 | beginning of βV3 | CTA CGT GGA ACA GTC TCC TTC | YVEQSP | 61 |
|  | R | MFC131 | end of βV3 | CAC GCG CAG AAA TAG ACA G | VYFCA | 57 |
| TcR βV5 | F | MFCb50 | beginning of βV5 | GTT CGG TGC TCT TTC TCT GC | MFGALSLH | 60 |
|  | R | MFCb54 | end of βV5 | GAC TGC AGT ATC AGT CGG CAC C | LVPTDTAV | 66 |
| TcR γV1 | F | MFCg56 | beginning of γV1 | GTC GCT GTA TTA CTG GCT CAT TG | MSLYYWL | 63 |
|  | R | MFCg59 | end of γV1 | GAG CGC ACA GTA ATA GGT GGC AG | TATYYCAL | 67 |
| TcR γV3 | F | MFCg58 | beginning of γV3 | GAA GGG TCA CGT CCT TGC G | MKGHVLA | 62 |
|  | R | MFCg61 | end of γV3 | GAT CCC AGA GTC ATC CTC | EDDSGI | 56 |
|  | F | MFC170 | beginning of γV3 | CAA TAA CCA GAG CAC CGG G | ITRAP | 56 |
|  | R | MFC171 | end of γV3 | AGA TCC CAG AGT CGT CCT C | EDDSGI | 56 |
| TcR δV3 | F | MFCd62 | beginning of δV3 | GAT TCC CCG TCC CTG GTG TC | DSPSLVS | 65 |
|  | R | MFCd66 | end of δV3 | CAG TGC ACA GTG ATA CAC AGC | AVYHCAL | 61 |
| TcR δV5 | F | MFCd63 | beginning of δV5 | GCA GCT ACT CAG TAT CTG G | MQLLSIW | 57 |
|  | R | MFCd67 | end of δV5 | GAA AGC ACA GTA ATA CAG AG | ALYYCAF | 54 |
| TcR δV7 | F | MFC172 | beginning of δV7 | CTG TCA CTC AGT TAT TCT CCT C | VTQLFS | 60 |
|  | R | MFC173 | end of δV7 | GCA GCC CAG TTA TAG TCA AAC | LTITGL | 60 |
| TcR δV12 | F | MFC174 | beginning of δV12 | CAG AGC CCA CCT CAG TTA C | QSPPQ; | 60 |
|  | R | MFC175 | end of δV12 | GAG CGC AGT AAT AGA TGG C | AIYYCA | 57 |
| TcR δV16 | R | MFC176 | end of δV16 | GCA GCT CCG AGA TAG ACA AC | LSISEL | 60 |
|  | F | MFC177 | beginning of δV16 | GAG TCC TGG CTC ACG CAA TC | ESWLTQ | 63 |
| TcR δV17 | F | MFC178 | beginning of δV17 | CAG TCT TGG TCA GAA ATA ACC | QSWSEIT | 57 |
|  | R | MFC179 | end of δV17 | CAA CTG AAG ATA AGT GAT CG | ITYLQL | 54 |

*Table 7 continued on next page*

*Table 7 continued*

| Primer | F/R | ID | Location | Nucleotide sequence (5' to 3') | Amino acid | Tm |
|--------|-----|-----|----------|-------------------------------|------------|-----|
| AID | F | MFC342 | beginning of AID exon 1 | AGG CAC GAG ACC TAC ATG TTG | RHETYML | 61 |
| | R | MFC347 | end of AID exon 2 | TGA ACC AGG TGA GGC GGT A | YRLTWF | 60 |
| B2M | F | MFC211 | first cysteine | AAC GTG TTG CTC TGT CAT GC | NVLLCHA | 58 |
| | R | MFC212 | before second cysteine | GGG GTG AAC TCC ACA TAA CG | RYVEFTP | 60 |

DOI: https://doi.org/10.7554/eLife.28477.023

## Sequence alignment and tree building

We used Geneious and BioEdit (v7.2.5, Ibis BioSciences, Carlsbad, CA) software to manage DNA sequence data. We aligned nucleotide and amino acid sequences using the ClustalW Multiple Alignment tool in Geneious with a gap penalty of 15, a gap extension penalty of 6.66, and free end gaps. We manually adjusted the alignments as necessary. We determined sequence relationships phylogenetically using the Geneious tree builder with default settings. We grouped sequences into unique V families based on 70% nucleotide sequence identity and 75% amino acid sequence identity (*Brodeur and Riblet, 1984*; *Rumfelt et al., 2004*) using the same α V numbering scheme as in *Criscitiello et al. (2010)*. We created graphical alignments in BioEdit and imported these files into Microsoft Word to generate figures. Our preliminary dataset contained 564 TcRα clones (encoding 286 unique amino acid sequences representing nine Vα families) from three tissues (PBL, spleen, thymus) of two sharks (*Joanie*, *Mary Junior*). Using this dataset, we separated sequences containing identical CDR3 rearrangements and counted mutations within each TcR 'clone family' bearing the V-J rearrangement from single founder thymocytes.

## Identification of TcR Vα genes in the nurse shark genome

We probed the filter sets for the *G. cirratum* BAC library (Arizona Genomics Institute, Tucson, AZ) of shark 'Yellow' and screened with variable segment and constant region probes for TcRα and TcRδ. We cultured several positive clones and isolated BAC DNA according to manufacturer's protocol with the Qiagen Large Construct Kit (Qiagen, Valencia, CA). We sent purified BAC DNA to the Duke University Center for Genomic and Computational Biology (Durham, NC) for PacBio SMRT (Menlo Park, CA) large insert (15–20 kb) library preparation, sequenced on the PacBio RSII platform with P6-C4 chemistry. Read correction and contig assembly were performed with the PBcR software (*Koren et al., 2012*), using the BLASR error correction method and the Celera Assembler 8.2. We annotated the resulting sequencing within the Geneious software suite (v9.1.5, Biomatters Ltd., Auckland, NZ) using a custom BLAST database of all TcR and IgH sequences for *G. cirratum* in the IMGT database (Montpellier, France).

Our search yielded 17 α/δ V germline segments, significantly fewer than expected based on TcR αV segment numbers in other species (*Murphy and Weaver, 2017*). Of these 17 segments, only 13 contained unique nucleotide sequences, and all V segments were highly similar to each other (69–100% nucleotide and 52–100% amino acid identity). Twelve germline V segments shared >93% nucleotide identity (>85% amino acid identity), with three segments differing by only a single nucleotide (*Figure 11*). Based on the variability we observed in our sequence data, these 17 germline α/δ V segments must represent only a small portion of the available Vs in the nurse shark genome.

We compared these 17 germline α/δ V segments to our TcR αV database containing all nine potential V families from two different sharks. All 17 germline α/δ V segments aligned to our αV4 data with >75% nucleotide identity, while 15 segments shared >93% nucleotide identity to at least one sequence in our αV4 dataset. Of the 60 sequences in our αV4 dataset, 37 sequences aligned specifically to eight germline α/δ V segments, with alignments containing one to 17 aV4 sequences per germline segment (nucleotide alignments shared >97% identity; *Figure 12.*). While we did observe nucleotide differences within alignments, most differed by fewer than four nucleotides from the germline α/δ V segments. Because several germline segments differed by only a single nucleotide and we are certain that we have not found all α/δ V segments in the genome, these differences could represent variation in alleles or individuals rather than mutation. Thus, we chose not to rely on these data for mutational analysis.

## Mutation frequency

We defined mutation frequency as the number of nucleotide changes divided by the total number of nucleotides within a particular region (e.g. FR, CDR, J, C) based on differences to a consensus sequence. We classified all nucleotide changes as either synonymous (SYN) or non-synonymous (NSYN) mutations based on whether or not the codon was unaltered or altered, respectively. For tandem base changes, we assessed the effect of each nucleotide change independent of its neighboring mutation(s). We then compared mutation frequencies between CDR and FR regions for all clone families that contained mutations using a Student's 1-tailed t-test unless otherwise noted.

## Determination of hotspots

We searched for the ProSite motifs DGYW/WRCH (G:C mutable target) and WA/TW (A:T mutable target) using the motif search function in Geneious. These motifs serve as common 'hotspots' for SHM within Ig variable regions, where AID favors the G/C bases within DGYW/WRCH motifs during the first phase of SHM while low-fidelity polymerases (i.e., polymerase η) preferentially target A/T bases within WA/TW motifs during the second phase of SHM (*Wei et al., 2015*; *Rogozin and Diaz,*

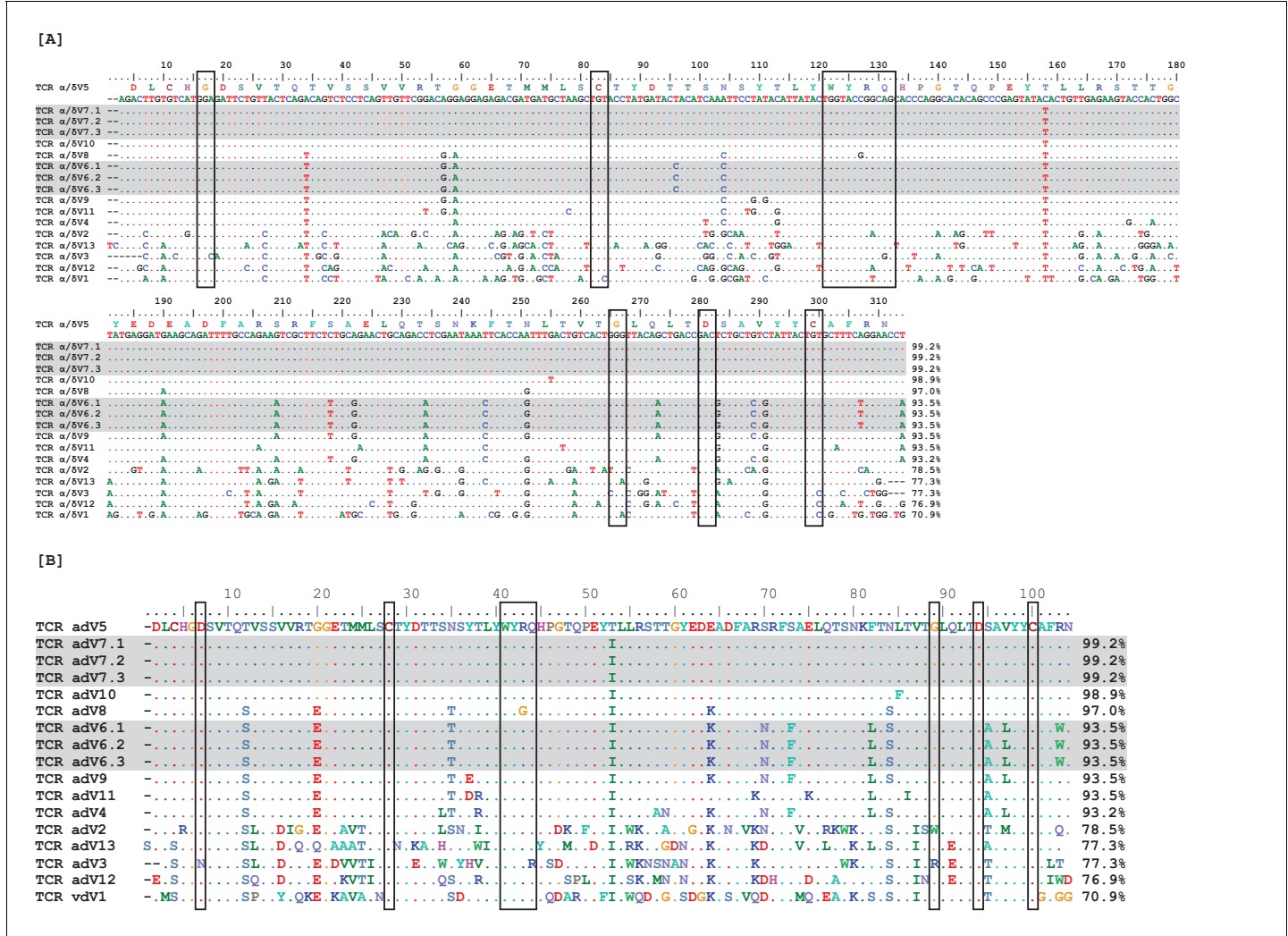

**Figure 11.** Observed TcR Alpha/Delta germline Vs exhibit high sequence identity. Nucleotide (**A**) and amino acid (**B**) alignments of 17 germline variable (V) region gene segments. Two V groups contained three identical germline gene segments each (highlighted in gray), leaving only 13 unique V gene segments. Boxes surround conserved amino acids. Numbers at the ends of sequences indicate percent identity to the first germline sequence (α/δ V5) within the alignment.

DOI: https://doi.org/10.7554/eLife.28477.024

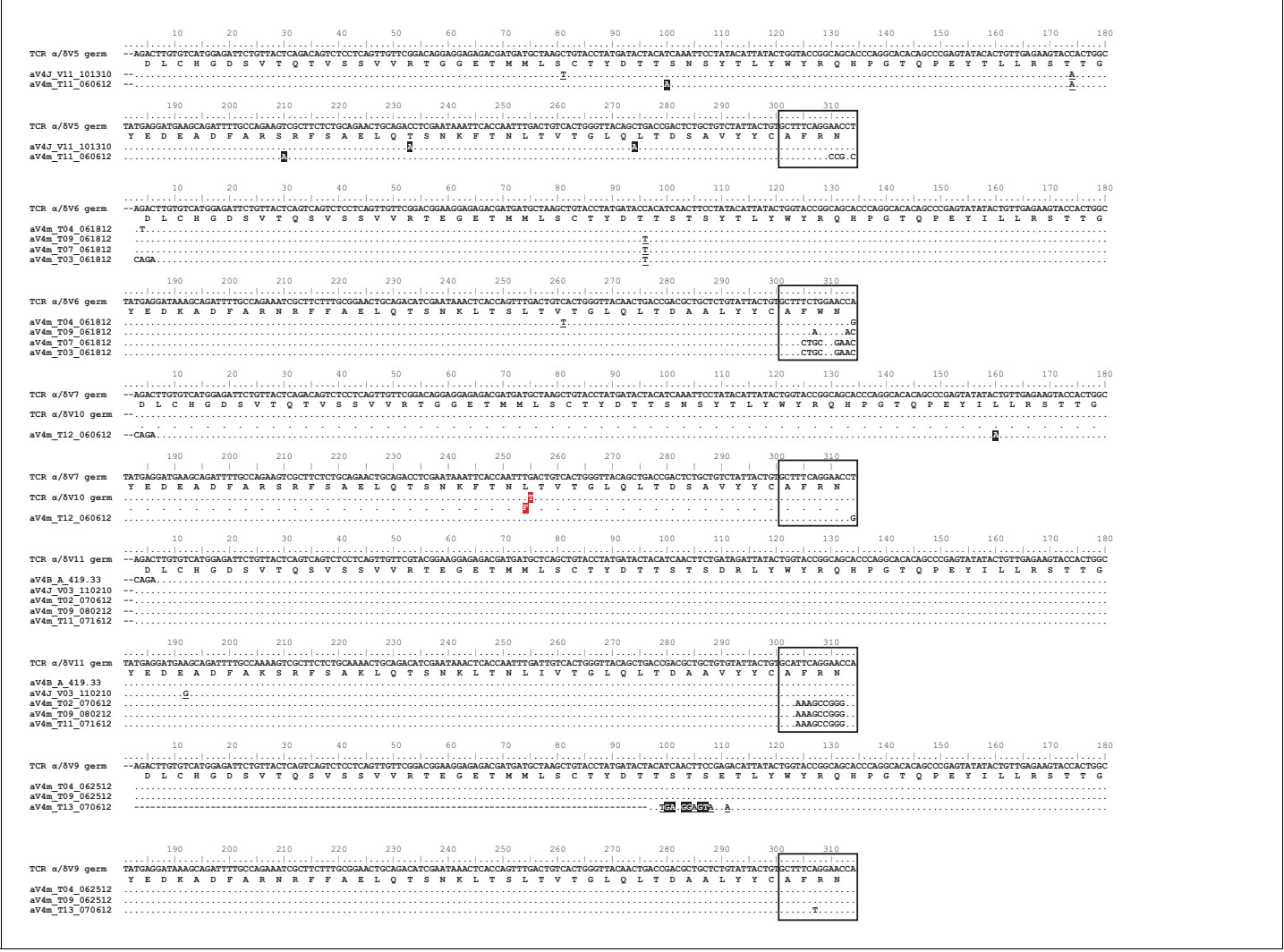

**Figure 12.** Observed germline sequences align only to TcR αV4 clones. Nucleotide alignments of TcR αV4 thymocyte clones to known germline V segments. Highlighted and underlined bases denote nonsynonymous and synonymous differences (respectively) to the germline V segment. Boxed regions represent nucleotides of the third complementarity-determining region (CDR3) according to IMGT guidelines, accounting for differences between clone and germline sequences. We highlight the single nucleotide/amino acid change between α/δ V7 and α/δ V10 germline segments in red.

DOI: https://doi.org/10.7554/eLife.28477.025

2004; *Chen et al., 2012*). We counted only motifs present in the consensus sequence (rather than those created by the mutation) as hotspots. For each domain, we first counted the number of target nucleotides within D<u>G</u>YW/WR<u>C</u>H or W<u>A</u>/<u>T</u>W hotspots. We then examined each mutation to determine if it occurred inside or outside a hotspot. We counted changes from the consensus sequence to a target nucleotide within its respective motif as a hotspot mutation. We defined the frequency of hotspot mutation as the number of mutations (of target nucleotides) occurring within hotspots divided by the total number of mutations for each region. From these data, we compared the mutability of bases between FR and CDR regions using $\chi^2$ Analysis.

## Base substitution indices

We calculated a mutability index for each nucleotide using methods similar to *Chen et al. (2012)*. In our case, we derived the expected number of mutations by multiplying the frequency of a particular nucleotide within a family of sequences (e.g. α V2) by the total number of observed mutations within that family. We then defined the mutability index as in *Chen et al. (2012)* [the observed number of mutations of a specific nucleotide divided by the expected number of mutations of that nucleotide,

with a value of 1.00 indicating random mutation]. We used $\chi^2$ analysis to compare mutability indices between FR and CDR regions.

## In situ hybridization

We used thymus tissue from an adult nurse shark for in situ hybridization as previously described (*Criscitiello et al., 2010*). We generated a probe for *G. cirratum* AID mRNA with primers NSAIDEH2 and NSAIDEH1 (*Table 7*) designed to amplify 211 base pairs of cDNA sequence of the first two AID exons (sequence and primers kindly shared by Ellen Hsu). We acquired images on an Axioscop2 microscope with AxioCam MRc5 (Zeiss, Thornwood CT) using Zeiss Axio Vision software.

Additionally, we performed fluorescence in situ hybridization (FISH) on adult nurse shark ('Black') thymus tissue. Slides contained two 8 µm thick sections of flash frozen thymus tissue preserved in OCT. We designed custom Stellaris FISH Probes against the TcR alpha constant region (αC) for T cell identification and exons 1 and 2 of *AID* by utilizing the Stellaris RNA FISH Probe Designer (Biosearch Technologies Inc., Petaluma, CA) available online at www.biosearchtech.com/stellarisdesigner (Version 2). We hybridized TcR αC with the CalFluor Red 610 fluorophore and the AID sequence with the Quasar 670 fluorophore for the Stellaris RNA FISH Probe set (Biosearch Technologies, Inc.). We followed all manufacturer's instructions for frozen tissue (available online at www.biosearchtech.com/stellarisprotocols), allowing hybridization probes to incubate for 16 hr. We counterstained slides with wash buffer containing 5 ng/mL of DAPI (Sigma-Aldrich, St. Louis, MO). We obtained 10x, 20x, and 63x images using a Zeiss Stallion Digital Imaging Workstation including a 2x CoolSnap HQ Camera and Zeiss Stallion software. We merged Z-stacked images of each fluorophore together and edited and processed images using ImageJ software, version 1.47 (*Schneider et al., 2012*).

## Real-time qPCR for AID expression

We synthesized cDNA from nurse shark spleen, thymus, muscle, and forebrain RNA (see RNA purification and extraction methods above) using SuperScript III First-strand Synthesis System (Thermo Fisher Scientific, Inc.) and a 1:1 mixture of oligo-dT and random hexamer primers. We then amplified cDNA using touchdown PCR on an MJ mini thermal cycler (Bio-Rad, Hercules, CA) and GoTaq colorless DNA polymerase (Promega Corp) using the following conditions: primary denaturing at 94°C for 2 min, five cycles at 94°C for 30 s and 56°C for 4 min; five cycles at 94°C for 30 s and 54°C for 4 min; 20 cycles at 94°C for 30 s, and 52°C for 30 s, and 72°C for 4 min; with a final extension at 72°C for 10 min. We visualized PCR products using agarose gel electrophoresis (as described above) to verify presence of AID in each tissue. We then cloned and sequenced the resulting PCR products to confirm the sequence was AID.

We looked for relative AID expression in shark spleen (positive control, where B cell AID-mediated SHM is known to occur), thymus, and forebrain (negative control) at four tissue concentrations (50 ng, 25 ng, 12.5 ng, and 6.25 ng) using the SYBR-green RT-PCR reagents kit (Thermo Fisher Scientific, Inc.) on a LightCycler 480 System (Roche Diagnostics Corp, Indianapolis, IN). We analyzed relative quantification using the LightCycler480 software and quantified relative AID expression using the ΔΔCq method. We normalized results against shark muscle tissue using beta2-microglobulin (β2M) as a reference gene. We present data as expression-fold changes of AID to β2M.

## Acknowledgements

We acknowledge expert technical assistance from Pat Chen and Ferenc Livak for helpful advice and comments on the manuscript. Ellen Hsu kindly shared a partial nurse shark AID sequence.

## Additional information

### Funding

| Funder | Grant reference number | Author |
| --- | --- | --- |
| National Science Foundation | IOS 1257829 | Michael F Criscitiello |
| National Institute of Allergy and Infectious Diseases | R01OD0549 | Martin F Flajnik |

The funders had no role in study design, data collection and interpretation, or the decision to submit the work for publication.

### Author contributions
Jeannine A Ott, Conceptualization, Data curation, Software, Formal analysis, Validation, Investigation, Visualization, Methodology, Writing—original draft; Caitlin D Castro, Yuko Ohta, Formal analysis, Investigation, Methodology; Thaddeus C Deiss, Software, Formal analysis, Investigation, Methodology, Writing—review and editing; Martin F Flajnik, Formal analysis, Funding acquisition, Investigation, Methodology, Project administration, Writing—review and editing; Michael F Criscitiello, Conceptualization, Formal analysis, Supervision, Funding acquisition, Investigation, Project administration, Writing—review and editing

### Author ORCIDs
Jeannine A Ott (iD) http://orcid.org/0000-0002-3537-8631
Michael F Criscitiello (iD) http://orcid.org/0000-0003-4262-7832

### Ethics
Animal experimentation: These studies were carried out in strict accordance with the recommendations in the Guide for the Care and Use of Laboratory Animals of the National Institutes of Health. The protocol was approved by the Animal Care and Use Committees at Texas A&M University and University of Maryland School of Medicine. Experiments in the Criscitiello lab were performed under Texas A&M University Institutional Biosafety Committee permit IBC 2014-293 and Animal Use Protocol 2015-0374.

### Decision letter and Author response
Decision letter https://doi.org/10.7554/eLife.28477.032
Author response https://doi.org/10.7554/eLife.28477.033

## Additional files

### Supplementary files
• Transparent reporting form
DOI: https://doi.org/10.7554/eLife.28477.026

### Data availability
T cell receptor sequences have been deposited in Genbank of NCBI. Alpha KY189332-KY189354 and KY366469-KY355487; Beta KY351708-KY366487; Gamma KY351639-KY351707; Delta KY346705-KY346816.

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
