## [Decision Letter]

Thank you for submitting your article "Somatic hypermutation of T cell receptor α chain contributes to thymic positive selection in nurse sharks thymus" for consideration by *eLife*. Your article has been favorably evaluated by Michel Nussenzweig (Senior Editor) and three reviewers, one of whom, David Schatz (Reviewer #1), served as a Guest Reviewing Editor. The reviewers have discussed the reviews with one another and the Reviewing Editor has drafted this decision to help you prepare a revised submission.

Summary:

The authors demonstrate and characterize somatic hypermutation (SHM) in TcRα, γ, and δ chains in the nurse shark. TcRα hypermutation is described for the first time in any TcRαβ system, although it was previously observed in sandbar shark TcRγδ chains. No mutation was observed for nurse shark TcRβ. SHM had been predicted to be disallowed in TcRαβ systems as it was never observed in mammals and has the potential to disrupt TcR-MHC I/II interactions. The findings here challenge that view and suggest that at least in shark, SHM of TcRα in the thymus contributes to repertoire diversification and potentially to α-β T cell positive selection. Consistent with the findings from TcR chain sequencing, the authors provide evidence in favor of AID expression in the thymic cortex and corticomedullary junction. The authors are experts in the field of comparative VDJ immunity and have published widely on this topic. The presentation of the findings and the discussion of their implications are clear and interesting. The finding highlights an important and unexpected aspect of TcR biology with implications for TcRαβ function in jawed vertebrates. This study should be of significant general interest to immunologists and evolutionary biologists.

Essential revisions:

1) The major challenge facing the authors is that they do not have the germline sequence of most nurse shark Vαα gene segments, and hence they have to infer a germline sequence from the cDNA sequences they obtain. To do this, the authors assume that if the same CDR3 sequence arises more than once, it must derive from the same original V-to-J rearrangement, and hence that sequence differences between two such clones must represent somatic mutations. Unfortunately, while this argument is strong for IgH, TcRδ, and TcRβ sequences that include D segments, it is potentially weak for TcRα rearrangements where a single joint is formed and junctional diversity can be low. To have confidence that duplicate CDR3 sequences represent unique rearrangements, one would want to examine the extent of nucleotide deletion and addition; only in joints with significant addition/deletion would one be confident that the junction is likely to be unique. The problem is compounded by the fact that VDJ recombination has a tendency to use microhomologies (if they exist) at the ends of the two gene segments to mediate joining, and this can lead to the same junction repeatedly. Was the same CDR3 sequence ever observed in different sharks? The authors need to justify their underlying assumption and provide an explicit discussion of the specificity of CDR3 sequence for rearrangements in each chain.

2) In addition, the data contain a significant puzzle: a very high level of shared CDR3 among a relatively small number of unique sequences. Specifically, 72 unique amino acid sequences, representing 12 V-α families, were obtained, which would correspond to an average of 6 unique sequences per V family, and an average of only 3 unique sequences per V family per shark. Given that the starting 5 microgram aliquots of RNA should derive from hundreds of thousands of cells, each starting RNA sample should contain a large number of independent V-J rearrangements for any given V. Given this and that there appear to be multiple Jα gene segments in the nurse shark, the chance of finding the same CDR3 sequence in a handful of sequences for a given V would be predicted to be very low. The chance of finding 42 sequences out of the 72 (more than half!) that share CDR3 with another sequence seems exceedingly unlikely. The authors need to carefully address this issue. The authors should provide a table that lists the number of unique aa sequences obtained for each V family within the 72 sequences, and of those, the number that have an identical CDR3.

The authors did in fact derive the germline sequence of two V family members (V4 and V9). Why not focus the analysis on these two families, where such ambiguities would be less of an issue? Furthermore, one of the reported set of sequences does arise from the V4 family, but it does not appear that the germline sequence information was taken advantage of. Why? And, were other sequences from V4 and V9 obtained in the dataset, and did they show mutations compared to the germline sequences?

It is noted that all reviewers agreed that the nature of the sequence differences observed (focus on hotspots, abundance of tandem mutations, etc.) are consistent with the mutations being real SHM events.

3) Expression of Aicda is not particularly clear in the ISH analysis. The data show a generally darker appearance with the antisense compared to the sense probe, but these differences are not completely convincing. Furthermore, it was difficult to distinguish the cortex vs. the medulla and the appearance is so uniform overall that claims for region specificity for Aicda are unconvincing. More annotation highlighting thymic structure and the probes used would be helpful. Scale bars should be included instead of objective magnification. Overall, it was felt that the Aicda ISH data need to be improved. Possibly other images or higher magnification images would help.

4) There were some concerns about the Aicda RT-PCR data. The results are expressed in delta Ct, which is a logarithmic measure, and it would be better to calculate a fold-difference in the expression of AID in spleen and thymus. How much of a difference in expression is there between the negative control (forebrain) and thymus? Was the resulting PCR product cloned and sequenced to confirm its identity? Are there B cells in the nurse shark thymus as there are in mouse and perhaps other cartilaginous fish such as the skate? If so, are the authors sure that AID expression is in T cells in the ISH and RT-PCR analyses? The authors should consider discussing this in the manuscript.

[Editors' note: further revisions were requested prior to acceptance, as described below.]

Thank you for resubmitting your work entitled "Somatic hypermutation of T cell receptor α chain contributes to positive selection in nurse shark thymus" for further consideration at *eLife*. Your article has been favorably evaluated by Michel Nussenzweig (Senior Editor) and three reviewers, one of whom served as Guest Reviewing Editor.

The manuscript has been improved but there are some remaining issues that need to be addressed before acceptance, as outlined below:

The authors have added new RNA fluorescence in situ hybridization data that strengthen the claim that AID is expressed in the shark thymus, and have adequately addressed some of the points previously raised. There remain a number of significant issues that need to be addressed satisfactorily before publication can be considered.

Major Points:

1) Based on the information provided by the authors in their response to the review, it is now clear that the analysis of mutations in TcRβ, γ, and δ (no CDR3 sequence obtained) was less rigorous and less definitive than in TcRαα (and, the TcRβ analysis is weaker than the rest because of the small number of sequences obtained (56)). Hence, all claims that mutations were not found in TcRβ must either be removed or carefully qualified. For example, the second sentence of the Abstract contrasts the findings at TcRβ (no mutation) and TcRα (mutation found) with no qualification, leaving the reader with the mistaken impression that the two analyses were equivalent and the two conclusions equally strong. The same problem was found in the last paragraph of the Introduction, subsection “Somatic hypermutation in TcR γV and TcR δV but not TcR βV”, and in the third paragraph of the Discussion.

2) "Since it is highly improbable that two thymocytes would generate identical CDR3 sequences during VJ recombination…" As was pointed out in the original review, this claim is not obvious; it should not be stated in this fashion. Indeed, there are a priori reasons to think that the same CDR3 sequence could readily arise from independent V-J rearrangements. The authors should present this claim as a hypothesis that they then carefully validate with data (which they begin to do in the first paragraph of the subsection “Somatic hypermutation in nurse shark TcR αV”). Furthermore, the conclusion of this sentence that "amplicons containing identical CDR3 sequences must have derived from the same progenitor" is overstated. At best, the conclusion could be that they are "likely to have derived from the same progenitor". A statistical analysis of the V-J sequences might allow the authors to make a quantitative claim of just how likely this is, but in the absence of such an analysis, the claims should be stated cautiously. The manuscript needs to be rewritten to accommodate these points. Same issue in the aforementioned paragraph.

3) Closely related to this: The authors have clarified that a larger number of sequences were obtained than was apparent in the initial submission, but this information is not provided in the Results section. Since most readers will not look at the Materials and methods section, it is suggested that the authors add information to the second paragraph of the subsection “Somatic hypermutation in nurse shark TcR αV” to inform readers of the total number of underlying sequences from the two relevant sharks. In addition, the Materials and methods states that sequences were obtained from four sharks; it is confusing that data from only two of the sharks was actually used. Why are all 4 sharks mentioned if data from only two of them were used? How many total sequences were obtained from the two relevant sharks? This is highly relevant given that these are (apparently) the only data from which the analyzed TcRα sequences are derived. This is directly relevant to the concern, expressed in the original review, that the identification of so many repeated CDR3 regions from a small sequence dataset is extremely unlikely. This issue has not been satisfactorily resolved in the resubmission. The requested table that shows how many unique aa sequences were obtained for each V family, and of those, the number that have an identical CDR3, was not provided. Nor have the authors provided a detailed description of the proportion of TcRα sequences that have N/P nucleotides (which would provide higher confidence of assigning SHM) versus those that are directly joined (less confidence). The reviewers remain concerned by the possibility that the number of repeat CDR3 sequences is exceedingly unlikely given the number of sequences obtained. The authors should carefully address this concern.

4) The new RNA fluorescence FISH analysis is a significant improvement over the ISH analysis, but despite the claimed specificity of the method used, experiments to document specificity are needed. Ideally, a tissue not expressing AID would be analyzed (or do the authors feel that the negative portions of the thymus provide sufficient specificity controls?). Given that the authors now present a detailed model for AID expression/SHM in shark thymus (Figure 10) based on the RNA fluorescence FISH data, it is particularly important to address specificity.

5) The claim made in the last paragraph of the subsection “Mutations, in situ hybridization, and AID expression in thymus “that TcRα expression is more diffuse than that of AID is unconvincing. TcRαA and AID expression appear to coincide in all areas visible in Figure 9. The authors should not push their interpretation just because it fits with the proposed role of AID in selection. Similarly, the organization of AID in rings around TcRα+ regions is intriguing and while the authors' interpretation in terms of relative timing of expression is plausible, it is quite speculative and should be clearly stated as such.

6) The in situ hybridization in Figure 8B (panels 3,4) for AID is still hard to interpret as the signal is weak and diffuse. The apparent expression pattern suggested by these images does not seem to correspond to the localized patterns seen in the fluorescent in situs in Figure 9 and Figure 9—figure supplement 1. Why is this? Is this a matter of scale? It is a concern that one set of data seems to contradict the other.

---

## [Author Response]

Essential revisions:1) The major challenge facing the authors is that they do not have the germline sequence of most nurse shark Vα gene segments, and hence they have to infer a germline sequence from the cDNA sequences they obtain. To do this, the authors assume that if the same CDR3 sequence arises more than once, it must derive from the same original V-to-J rearrangement, and hence that sequence differences between two such clones must represent somatic mutations. Unfortunately, while this argument is strong for IgH, TcRδ, and TcRβ sequences that include D segments, it is potentially weak for TcRα rearrangements where a single joint is formed and junctional diversity can be low. To have confidence that duplicate CDR3 sequences represent unique rearrangements, one would want to examine the extent of nucleotide deletion and addition; only in joints with significant addition/deletion would one be confident that the junction is likely to be unique. The problem is compounded by the fact that VDJ recombination has a tendency to use microhomologies (if they exist) at the ends of the two gene segments to mediate joining, and this can lead to the same junction repeatedly. Was the same CDR3 sequence ever observed in different sharks? The authors need to justify their underlying assumption and provide an explicit discussion of the specificity of CDR3 sequence for rearrangements in each chain.

In our shark sequences, we found substantial diversity within CDR3 regions. We present data to illustrate CDR3 junctional diversity and include a new figure (Figure 4) depicting this diversity. (We observed similar results in other α V segments as well.) We never observed the same CDR3 sequence in more than one shark.

We included a paragraph under the “Somatic hypermutation in nurse shark TcR αV” section of the results to discuss these data: “Despite the absence of diversifying segments, α CDR3s exhibited substantial variation within our shark sequences. For example, TcRα V1 sequences using the same V and J segments had CDR3 lengths that differed by as many as 6 amino acids (12 nucleotides), and few CDR3s shared more than one amino acid within this V-J join (Figure 4). Further, we never observed the same CDR3 sequence in more than one shark, suggesting both exonuclease activity and addition of N and P nucleotides help diversify α CDR3s in nurse shark.”

2) In addition, the data contain a significant puzzle: a very high level of shared CDR3 among a relatively small number of unique sequences. Specifically, 72 unique amino acid sequences, representing 12 V-α families, were obtained, which would correspond to an average of 6 unique sequences per V family, and an average of only 3 unique sequences per V family per shark. Given that the starting 5 microgram aliquots of RNA should derive from hundreds of thousands of cells, each starting RNA sample should contain a large number of independent V-J rearrangements for any given V. Given this and that there appear to be multiple Jα gene segments in the nurse shark, the chance of finding the same CDR3 sequence in a handful of sequences for a given V would be predicted to be very low. The chance of finding 42 sequences out of the 72 (more than half!) that share CDR3 with another sequence seems exceedingly unlikely. The authors need to carefully address this issue. The authors should provide a table that lists the number of unique aa sequences obtained for each V family within the 72 sequences, and of those, the number that have an identical CDR3.

As written, the paper does suggest an incredible coincidence of shared CDR3s among our sequences. However, we actually acquired significantly more sequences than the 72 we describe in the paper. We have revised this section to include a discussion of the total number of sequences (564) from which we found those with shared CDR3s.

Within the Materials and methods section titled “Sequence Alignment and Tree Building,” we added: “Our preliminary dataset contained 564 TcRα clones (encoding 286 unique amino acid sequences representing nine Vα families) from three tissues (PBL, spleen, thymus) of four sharks (Joanie, Mary Junior, Yellow, and 1299). Using this dataset, we separated sequences containing identical CDR3 rearrangements and counted mutations within each TcR “clone family” bearing the V-J rearrangement from single founder thymocytes.”

The authors did in fact derive the germline sequence of two V family members (V4 and V9). Why not focus the analysis on these two families, where such ambiguities would be less of an issue? Furthermore, one of the reported set of sequences does arise from the V4 family, but it does not appear that the germline sequence information was taken advantage of. Why? And, were other sequences from V4 and V9 obtained in the dataset, and did they show mutations compared to the germline sequences?

We added two figures and a couple of paragraphs in the Materials and methods section “Identification of TcR Genes in the Nurse Shark Genome” explaining our reasons for not using germline alignments for SHM analysis:

“Our search yielded 17 α/δ V germline segments, significantly fewer than expected based on TcR αV segment numbers in other species (Murphy and Weaver, 2017). […] Thus, we chose not to rely on these data for mutational analysis.”

It is noted that all reviewers agreed that the nature of the sequence differences observed (focus on hotspots, abundance of tandem mutations, etc.) are consistent with the mutations being real SHM events.3) Expression of Aicda is not particularly clear in the ISH analysis. The data show a generally darker appearance with the antisense compared to the sense probe, but these differences are not completely convincing. Furthermore, it was difficult to distinguish the cortex vs. the medulla and the appearance is so uniform overall that claims for region specificity for Aicda are unconvincing. More annotation highlighting thymic structure and the probes used would be helpful. Scale bars should be included instead of objective magnification. Overall, it was felt that the Aicda ISH data need to be improved. Possibly other images or higher magnification images would help.

We present new fluorescence in situ hybridization (FISH) data to illustrate more clearly the thymic architecture and regional specificity of AID expression within the shark thymus. The development of this technique delayed our resubmission. In a new figure (Figure 9), we provide a composite of seven Z-stacked images (10x) depicting the localization of AID expression to the cortico-medullary junction and medullary regions of shark thymus. We also present individual fluorophore images of DAPI, TcRα, and AID and the Z-stacked merged images (10x, 20x, and 63x) to illustrate the pattern of AID and TcRα expression within four locations of shark thymus. To ensure our probes were hybridizing independently of one another, we probed AID and TRA individually on adjacent slides and looked for co-localization of the probes (see Figure 9—figure supplement 1).

We also included a paragraph in the Results section describing these data: “We confirmed our colorimetric ISH results with RNA fluorescence in situ hybridization (FISH) using probes against the TcR α constant region (αC) and exons 2 and 3 of AID. […] These daughter T cells then express AID, promoting somatic hypermutation within their TcR α sequences during a time when cells also undergo positive and negative selection.”

4) There were some concerns about the Aicda RT-PCR data. The results are expressed in delta Ct, which is a logarithmic measure, and it would be better to calculate a fold-difference in the expression of AID in spleen and thymus. How much of a difference in expression is there between the negative control (forebrain) and thymus?

We present expression-fold changes in the different tissues (see Figure 8A). We also included the expression-fold difference between forebrain and thymus: “We confirmed the expression of AID in the thymus through real-time RT-qPCR, where thymus tissue expressed AID at roughly half (0.7x) the levels found in spleen (where B cell SHM is known to occur) and nearly 6x the levels observed in forebrain (negative control; Figure 8A).”

Was the resulting PCR product cloned and sequenced to confirm its identity?

Yes. We included a sentence in the qPCR Materials and methods section to inform readers: “We then cloned and sequenced the resulting PCR products to confirm the sequence was AID.”

Are there B cells in the nurse shark thymus as there are in mouse and perhaps other cartilaginous fish such as the skate? If so, are the authors sure that AID expression is in T cells in the ISH and RT-PCR analyses? The authors should consider discussing this in the manuscript.

We addressed thymic B cells in the Discussion: “Although recent studies report B cells present in the thymus of mice (Perera and Huang, 2015, Perera et al., 2013), Miracle et al. (2001) examined B cell expression levels (IgM and IgX) in various tissues (including thymus) at different ages in the clearnose skate. […] Further, preliminary results using FISH probes to IgM constant region also indicate that adult nurse sharks do not express IgM in the thymus."

These preliminary data (both RT-qPCR and FISH) give us confidence that the phenomenon we are seeing is not due to the presence of B cells in the thymus. However, these data are part of a larger study with a different aim, and we prefer not to report the data in this paper. However, we provide them here for you to assess their potential importance. The bar graph (Author response image 1) illustrates RT-qPCR expression of various immune cell markers in thymus tissue from sharks of varying ages. The red bars indicate IgM/IgW expression, which is clearly absent in the older sharks (red box).

Using FISH (Author response image 2), we found many IgM transcripts expressed in spleen (right) but few expressed in thymus (right). With the lack of IgM expression indicated by RT-qPCR, this may suggest that cells transcribe the IgM message when the DNA is open and the cell also is transcribing TcR genes, but translation of the IgM message does not occur. We need more evidence to make further conclusions.

**Author response image 2. respfig2:** 

[Editors' note: further revisions were requested prior to acceptance, as described below.]

Major Points:1) Based on the information provided by the authors in their response to the review, it is now clear that the analysis of mutations in TcRβ, γ, and δ (no CDR3 sequence obtained) was less rigorous and less definitive than in TcRα (and, the TcRβ analysis is weaker than the rest because of the small number of sequences obtained (56)). Hence, all claims that mutations were not found in TcRβ must either be removed or carefully qualified. For example, the second sentence of the Abstract contrasts the findings at TcRβ (no mutation) and TcRα (mutation found) with no qualification, leaving the reader with the mistaken impression that the two analyses were equivalent and the two conclusions equally strong. The same problem was found in the last paragraph of the Introduction, subsection “Somatic hypermutation in TcR γV and TcR δV but not TcR βV”, and in the third paragraph of the Discussion.

Agreed. We are in the process of repeating the mutation analysis of the other chains to the same rigor as α. Collecting enough CDR3-sharing sequences is taking some time. However, for this manuscript, we removed language that implies the analyses or conclusions are equivalent. In the above order, these changes are as follows:

“Remarkably, we found SHM acting in the thymus on α chain of shark T cell receptors (TcR).”

“Our data suggest that SHM of TcRα is involved in primary T cell repertoire diversification and the enhancement of positive selection in the thymic cortex.”

“Somatic hypermutation in TcR γV and TcR δV”.

“Remarkably we also detected SHM occurring in the α chain of αβ T cells.”

2) "Since it is highly improbable that two thymocytes would generate identical CDR3 sequences during VJ recombination…" As was pointed out in the original review, this claim is not obvious; it should not be stated in this fashion. Indeed, there are a priori reasons to think that the same CDR3 sequence could readily arise from independent V-J rearrangements. The authors should present this claim as a hypothesis that they then carefully validate with data (which they begin to do in the first paragraph of the subsection “Somatic hypermutation in nurse shark TcR αV”). Furthermore, the conclusion of this sentence that "amplicons containing identical CDR3 sequences must have derived from the same progenitor" is overstated. At best, the conclusion could be that they are "likely to have derived from the same progenitor". A statistical analysis of the V-J sequences might allow the authors to make a quantitative claim of just how likely this is, but in the absence of such an analysis, the claims should be stated cautiously. The manuscript needs to be rewritten to accommodate these points. Same issue in the aforementioned paragraph.

We modified this section to be less speculative. “Since it is highly unlikely that two thymocytes would generate identical CDR3 sequences during VJ recombination, we predict that amplicons containing identical CDR3 sequences derived from the same progenitor and thus must contain the same germline V and J segments. […] Therefore, even in the absence of an assembled locus, we were able to evaluate mutation to germline αV segments by considering changes within only those thymocyte clones containing identical CDR3s.”

3) Closely related to this: The authors have clarified that a larger number of sequences were obtained than was apparent in the initial submission, but this information is not provided in the Results section. Since most readers will not look at the Materials and methods section, it is suggested that the authors add information to the second paragraph of the subsection “Somatic hypermutation in nurse shark TcR αV” to inform readers of the total number of underlying sequences from the two relevant sharks. In addition, the Materials and methods states that sequences were obtained from four sharks; it is confusing that data from only two of the sharks was actually used. Why are all 4 sharks mentioned if data from only two of them were used? How many total sequences were obtained from the two relevant sharks? This is highly relevant given that these are (apparently) the only data from which the analyzed TcRα sequences are derived. This is directly relevant to the concern, expressed in the original review, that the identification of so many repeated CDR3 regions from a small sequence dataset is extremely unlikely. This issue has not been satisfactorily resolved in the resubmission. The requested table that shows how many unique aa sequences were obtained for each V family, and of those, the number that have an identical CDR3, was not provided. Nor have the authors provided a detailed description of the proportion of TcRα sequences that have N/P nucleotides (which would provide higher confidence of assigning SHM) versus those that are directly joined (less confidence). The reviewers remain concerned by the possibility that the number of repeat CDR3 sequences is exceedingly unlikely given the number of sequences obtained. The authors should carefully address this concern.

We apologize for not including the requested table. We misunderstood the initial request. We include this table now (Table 1) and discuss the table in the Results section titled “Somatic hypermutation in nurse shark TcR αV”:

“Our preliminary V α dataset contained 539 TcRα clones (encoding 286 unique amino acid sequences representing nine V α families) from three tissues (PBL, spleen, thymus) of two sharks (Joanie, Mary Junior). […] We include these 45 sequences in our TcR Vα dataset (see Table 1 – source data for sequence data).”

During the construction of this new table, we realized we had omitted sequences in our initial mutation analysis. Further, we decided to divide our aV4 group so that we estimated mutation even more conservatively. Thus, we reanalyzed our entire mutation data set. While the overall conclusions of this study did not change, some of the results did change. Most notably, we found that A:T mutations did not prefer WA/TW hotspots. This is consistent with data reported in sandbar sharks for TcR γ (Chen et al., 2012). However, our data suggest a different A:T motif may be targeted or that sharks employ a different mechanism than DNA pol eta use during mismatch repair to alter A:T nucleotides.

“A:T nucleotides did not appear to prefer WA/TW hotspots (p=0.2248), though they were 2.3x as likely to mutate within hotspots than outside hotspots. […] This result suggests that an alternate A:T motif is targeted or that sharks employ another mechanism altogether to alter A:T nucleotides.”

Further, we no longer found a significant difference in tandemly mutated bases between CDRs and FRs, which is consistent with results observed for TcR-γ in sandbar sharks. “Finally, though we found more tandemly mutated bases in CDRs (41 of 81, or 50.6% of all CDR mutations) than in FRs (73 of 192, or 38.0% of all FR mutations), this difference was not significant; p=0.721; Table 2).”

We also added a section at the beginning of the Materials and methods to clarify the confusion over sharks. We used sequences from only two sharks in our mutation analyses. However, we used published data from two different sharks to aid in loci identification to ensure consistency. This paragraph reads as follows: “Study Animals: TCR sequence data used in this study came from two adult female nurse sharks (“Joanie” and “Mary Junior”; *Ginglymostoma cirratum*) delivered by caesarian section off the Florida Keys and matured in the aquatic vivarium of the University of Maryland’s Center of Marine Biotechnology. We used published T cell αV sequences (Criscitiello et al., 2010) as a baseline for αV locus numbering (sharks Yellow and 1299), though we did not analyze any of these sequences for mutation.”

As for the presence of N/P nucleotides, please see our response to #2, above.

4) The new RNA fluorescence FISH analysis is a significant improvement over the ISH analysis, but despite the claimed specificity of the method used, experiments to document specificity are needed. Ideally, a tissue not expressing AID would be analyzed (or do the authors feel that the negative portions of the thymus provide sufficient specificity controls?). Given that the authors now present a detailed model for AID expression/SHM in shark thymus (Figure 10) based on the RNA fluorescence FISH data, it is particularly important to address specificity.

We analyzed a non-immune tissue (shark brain) as a negative FISH control for AID and TcRα fluorescence. We created a new supplementary figure with images of these slides (Figure 9—figure supplement 2). This figure depicts some background fluorescence regardless of probe application. However, we observed no AID or TcRα expression in brain tissue like the patterns we saw in thymus (immune tissue).

5) The claim made in the last paragraph of the subsection “Mutations, in situ hybridization, and AID expression in thymus “that TcRα expression is more diffuse than that of AID is unconvincing. TcRαA and AID expression appear to coincide in all areas visible in Figure 9. The authors should not push their interpretation just because it fits with the proposed role of AID in selection. Similarly, the organization of AID in rings around TcRα+ regions is intriguing and while the authors' interpretation in terms of relative timing of expression is plausible, it is quite speculative and should be clearly stated as such.

We modified the text to better reflect what our images depict (see also #7, below). “Our FISH results indicated a more specific TcR αC signal within the inner cortex and medulla adjacent to the CMJ (Figure 9; Figure 9—figure supplement 1), regions where, in mammals, cells actively rearrange their α chain and where mature αβ T cells are found. […] Thus, we observed a consistent pattern of expression where a “ring” of cells expressing both TcR αC and AID surround a central cell expressing only TcR αC.”

6) The in situ hybridization in Figure 8B (panels 3,4) for AID is still hard to interpret as the signal is weak and diffuse. The apparent expression pattern suggested by these images does not seem to correspond to the localized patterns seen in the fluorescent in situs in Figure 9 and Figure 9—figure supplement 1. Why is this? Is this a matter of scale? It is a concern that one set of data seems to contradict the other.

We auto-adjusted the colors on the CISH image in ImageJ to darken the signal expression [old image (L) and darkened image (R), Author response image 3]. We adjusted images in all eight panels uniformly, ensuring we did not alter expression differences between panels. This should make it easier to see the signal itself, with AID expression localized to the cortex and CMJ and minimal expression within the medulla.

**Author response image 3. respfig3:** 

As you point out, it is a much more diffuse signal. Because the Stellaris FISH method uses a mixture of short probes for binding the transcript, the pattern we see in the is much more specific – the image will only show those areas with many bound probes rather than the presence of a transcript in general. For example, the CISH image of TcR αC from Criscitiello et al.,2010 (Author response image 4) shows a much more diffuse pattern of expression than what we see with FISH as well.

**Author response image 4. respfig4:** 

We believe the expression patterns revealed with the two different methods are both correct, but they tell us different things. AID (and TcR αC) expression occurs throughout the cortex (and medulla for TcR αC; based on the diffuse signal from CISH), but the greatest expression of both messages occurs near the CMJ (based on the much more precise and higher intensity signal from FISH). We currently are working with another method that incorporates both a colorimetric and fluorescent signal to verify this explanation, but we have not been successful with that approach yet.

We modified the text to explain this difference in method.

CISH section: “Further, colorimetric in situ hybridization (CISH) of nurse shark thymus revealed a diffuse signal of TcR β (Figure 8B, panels 1,2) and AID (Figure 8B, panels 3,4) mRNA expression throughout the thymic cortex. However, AID expression was greatest in the central cortex and cortico-medullary junction (CMJ), while TcR β expression was highest in the outer cortex.”

FISH section: “We refined our CISH results with RNA fluorescence in situ hybridization (FISH) using probes against the TcR α constant region (αC) and exons 2 and 3 of AID. […] These daughter T cells then express AID (and TcR αC), promoting somatic hypermutation within their TcR α sequences during a time when cells also undergo positive and negative selection.”